# The role of lattice dynamics in ferroelectric switching

Qiwu Shi [1,2,13✉], Eric Parsonnet [3,13], Xiaoxing Cheng[4,13], Natalya Fedorova[5,13], Ren-Ci Peng[6,11], Abel Fernandez[1], Alexander Qualls [3], Xiaoxi Huang[1], Xue Chang[2], Hongrui Zhang[1], David Pesquera [1], Sujit Das [1,12], Dmitri Nikonov [7], Ian Young [7], Long-Qing Chen [4], Lane W. Martin [1,8], Yen-Lin Huang [1,9✉], Jorge Íñiguez [5,10] & Ramamoorthy Ramesh [1,3,8✉]

Reducing the switching energy of ferroelectric thin films remains an important goal in the pursuit of ultralow-power ferroelectric memory and logic devices. Here, we elucidate the fundamental role of lattice dynamics in ferroelectric switching by studying both freestanding bismuth ferrite ($BiFeO_3$) membranes and films clamped to a substrate. We observe a distinct evolution of the ferroelectric domain pattern, from striped, 71° ferroelastic domains (spacing of ~100 nm) in clamped $BiFeO_3$ films, to large (10's of micrometers) 180° domains in freestanding films. By removing the constraints imposed by mechanical clamping from the substrate, we can realize a ~40% reduction of the switching voltage and a consequent ~60% improvement in the switching speed. Our findings highlight the importance of a dynamic clamping process occurring during switching, which impacts strain, ferroelectric, and ferro-distortive order parameters and plays a critical role in setting the energetics and dynamics of ferroelectric switching.

[1] Department of Materials Science and Engineering, University of California, Berkeley, CA 94720, USA. [2] College of Materials Science and Engineering, Sichuan University, Chengdu 610065, China. [3] Department of Physics, University of California, Berkeley, CA 94720, USA. [4] Department of Materials Science and Engineering, Penn State University, University Park, Pennsylvania 16802 PA, USA. [5] Materials Research and Technology Department, Luxembourg Institute of Science and Technology, L-4362 Esch/Alzette, Luxembourg. [6] Electronic Materials Research Laboratory, Key Laboratory of the Ministry of Education & International Center for Dielectric Research, School of Electronic Information and Engineering, Xi'an Jiaotong University, 710049 Xi'an, China. [7] Components Research, Intel Corporation, Hillsboro, OR 97142, USA. [8] Materials Sciences Division, Lawrence Berkeley National Laboratory, Berkeley, CA 94720, USA. [9] Department of Materials Science and Engineering, National Yang Ming Chiao Tung University, Hsinchu 30010, Taiwan. [10] Department of Physics and Materials Science, University of Luxembourg, L-4422 Belvaux, Luxembourg. [11] Present address: School of Advanced Materials and Nanotechnology, Xidian University, Xi'an 710126, China. [12] Present address: Material Research Centre, Indian Institute of Science, Bangalore 560012, India. [13] These authors contributed equally: Qiwu Shi, Eric Parsonnet, Xiaoxing Cheng, Natalya Fedorova. ✉email: shiqiwu@scu.edu.cn; yenlin.jack.huang@gmail.com; rramesh@berkeley.edu

The last three decades have witnessed a significant interest in the science and technology of ferroelectric thin films, driven by the fascinating fundamental physics of the polar state in reduced dimensions as well as the technological applications, for example, in nonvolatile memories[1,2], which utilize the switchable nature of the ferroelectric polarization to store information. In the case of proper ferroelectrics[3–6], where the spontaneous polarization arises from the freezing of a soft-phonon mode at the Curie temperature, the coupling between the dipolar order and the lattice is much stronger than, for example, the coupling of spins to the lattice in ferromagnets[7–9]. Focusing on proper ferroelectrics such as $BiFeO_3$ (BFO), $PbTiO_3$ (PTO), and $BaTiO_3$ (BTO), this strong coupling between the lattice and the spontaneous electric dipoles means that switching of the polar state is accompanied by a corresponding, dynamic lattice distortion during switching. It is widely believed that the fundamental limit on ferroelectric switching speed is thus set by the phonon-dispersion relation, and specifically the group velocity for acoustic phonons in the system (i.e., speed of sound), which sets a limit on how fast the lattice can respond. For films clamped to a substrate, the substrate will undoubtedly modify the phonons of the thin film, altering their energetics and dispersion characteristics. For example, a perovskite substrate that does not contain any oxygen-octahedral tilts will probably soften (and reduce the equilibrium amplitude of) the $O_6$ tilts (ferrodistortive order parameter) of a BFO film, which will in turn impact, and likely facilitate, ferroelectric switching. In contrast, a perovskite substrate presenting rigid oxygen octahedral tilts may harden the corresponding phonons of the film, and may act as a "built-in field" of sorts for the BFO tilts; this will most likely result in slower switching, and potentially even modify the switching pathway. Clearly, the substrate, and the mechanical boundary conditions it imposes, plays a critical role in influencing the lattice dynamics of the film.

Even if we restrict ourselves to simple considerations such as those just mentioned—ignoring subtle effects related to phonon dynamics and their interplay with polarization and strains—it is clear that this is an exceedingly difficult problem. As such, we devise a tractable set of theoretical calculations and experiments that aims to answer a question that addresses how lattice dynamics influence polarization reversal, namely, what is the role of the substrate in influencing ferroelectric switching? We begin by considering the clamping effect, or resistance to structural deformation, which imposes an additional energy barrier that must be overcome to induce switching in films constrained to a substrate. Such an enhanced energy barrier can be understood as a coupling between the dynamic lattice strain and the primary order parameters in the system, which manifests itself both in the energy required to switch the state (coercive field) and in the switching time (both nucleation and growth regimes of polarization reversal). Indeed, previous work combining phase-field modeling with in situ-biasing transmission electron microscopy to study mechanical and electrical loading of relaxor ferroelectrics has demonstrated the importance of such mechanical constraints in establishing ferroelastic switching energies[10,11]. Here, we present a detailed theoretical and experimental analysis of the role of substrate clamping in influencing ferroelectric switching in the proper ferroelectric/multiferroic BFO. While all thin-film ferroelectrics are subject to clamping constraints from the substrate, it can play a larger role in inhibiting ferroelastic switching pathways[12,13]. BFO, which follows a two-step polarization-switching pathway (consisting of out-of-plane (109°) and in-plane (71°) steps, with its ferrodistortive oxygen octahedral tilts following the ferroelectric polarization[14,15]), is therefore an ideal candidate for studying the role of clamping in impacting the switching of coupled primary order parameters. Previous

theoretical works have developed highly successful theories for the equilibrium energetics of the BFO system, including the effects from oxygen octahedral tilting[16,17], though they have not addressed how substrate clamping influences such energetics (or dynamics) of the switching process. We theoretically study varying degrees of clamping, and introduce the notion of "strain + tilt clamping" where the substrate influences not only the ferroelectric and strain order parameters, but also, importantly, the oxygen octahedral tilts. We show that "strain clamping" alone (ignoring the role of the substrate in clamping the ferrodistortive order) is insufficient to explain the changes to the energetics and dynamics of switching in freestanding vs clamped BFO films, which we observe in our experiments. Interestingly, phase field calculations reveal that the rotation of the ferrodistortive order slows the switching by almost an order of magnitude in comparison to the case where oxygen octahedral tilts are ignored. Our findings are further complemented by similar measurements in the literature[18] on thin films of ferroelectric BTO, a prototypical tetragonal ferroelectric, a finding that highlights how substrate clamping impacts a variety of ferroelectric materials. In the case of BTO[18] and in the present study, the data reveal a clear impact on the switching voltage (a measure of the barrier energy) as well as the switching dynamics (as manifested by changes in the switching time). These observations indicate that the effects from mechanical clamping by the substrate are broadly applicable to all displacive ferroelectrics. Such an understanding is essential, as the 100 mV switching-voltage goal remains a grand challenge for the field[1,2].

## Results

While there have been a large number of studies of quasi-static switching behavior and equilibrium properties of thin films[19–23], there have been fewer studies of the limits and timescales of fast switching[24–28], and even fewer on the role of substrate-clamping effects in influencing ferroelectric switching below 1 μs[18]. A key question is how to quantify the role of the substrate in dictating the switching process and whether switching can be studied experimentally without the influence of the substrate. Freestanding ferroelectric membranes have recently emerged as an exciting platform to study the role of mechanical constraints in ferroelectric systems[29], and here, we attempt to quantitatively address the effect of mechanical clamping by using a combination of thermodynamic calculations, phase-field simulations, piezoresponse force microscopy, and quasi-static and dynamic switching measurements on epitaxial, substrate-attached, and freestanding versions of the same thin films (Fig. 1a).

To quantitatively understand the switching-energy landscape with and without substrate clamping in BFO (Fig. 1b), we modeled the thermodynamic free energy within the context of the Landau theory for ferroelectrics[30–32], using a potential of the form[33]

$$f = \alpha_{ij}p_ip_j + \alpha_{ijkl}p_ip_jp_kp_l + \beta_{ij}\theta_i\theta_j + \beta_{ijkl}\theta_i\theta_j\theta_k\theta_l \\ + t_{ijkl}p_ip_j\theta_k\theta_l + \frac{1}{2}C_{ijkl}(\epsilon_{ij} - \epsilon_{ij}^0)(\epsilon_{kl} - \epsilon_{kl}^0) \quad (1)$$

where $p_i$, $\theta_i$, and $\epsilon_{ij}$ refer to the ferroelectric polarization, ferrodistortive rotation of the oxygen octahedra, and strain, respectively, while $\epsilon_{ij}^0 = \lambda_{ijkl}\theta_k\theta_l + Q_{ijkl}p_kp_l$. Additional details are provided in "Methods" and Supporting Information Section 1, 2. We use two sets of parameters for this Landau potential: a first set directly fitted to first-principles results (Fedorova N., Nikonov D., Young I. & Íñiguez J., First-principles Landau-like potential for $BiFeO_3$ and related materials. Unpublished, 2022) (nominally at 0 K) and a second one corresponding to the room-temperature Ginzburg–Landau potential for BFO previously introduced in[33], the latter of which is the same model used for the phase-field simulations of ferroelectric switching

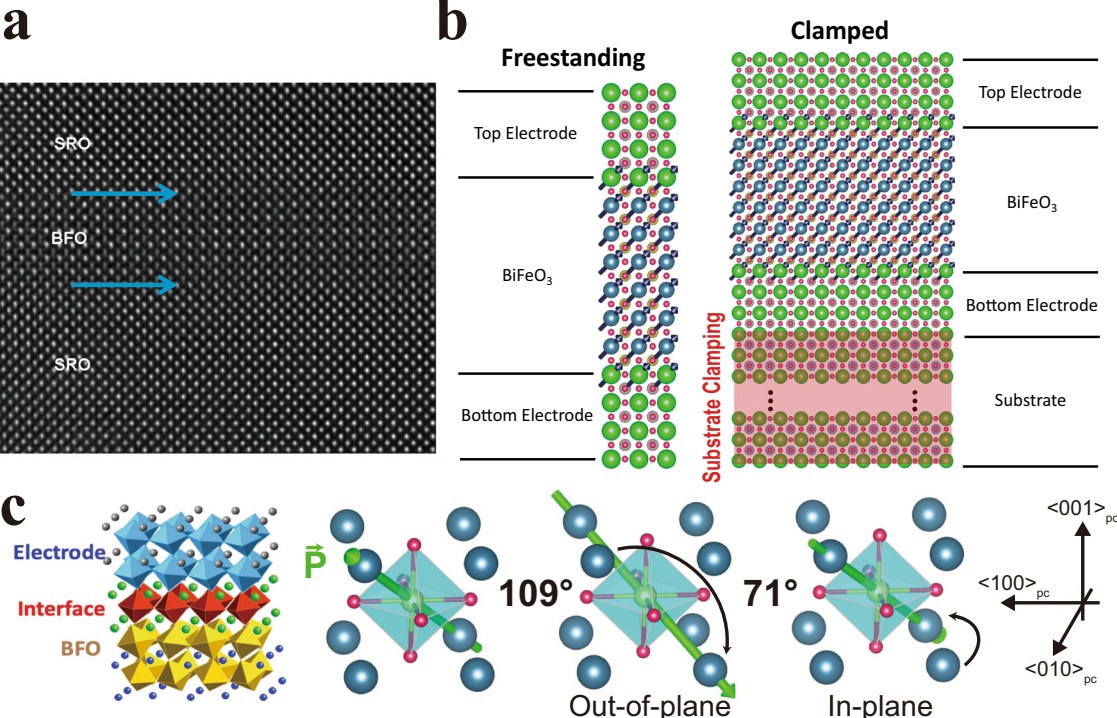

**Fig. 1 Role of clamping. a** Transmission electron microscope (TEM) image of SRO/BFO/SRO heterostructure. **b** Schematic highlighting significant mechanical constraints imposed by the substrate compared with the freestanding film. **c** SRO/BFO interface schematic showing ferrodistortive oxygen octahedra rotations and switching pathway (109° out-of-plane followed by 71° in-plane) for BFO films.

discussed below. Guided by previous literature[14,34], we know that upon the application of an out-of-plane electric field, BFO typically undergoes 180° switching via a two-step process (Fig. 1c): a 109° switch (where the out-of-plane polarization component reverses together with one in-plane component) followed by a 71° switch (where the remaining in-plane component reverses), or vice versa. In our thermodynamic analyses, we calculate the free-energy profile associated with the two switching steps while considering different levels of clamping.

First, we calculate the case of no clamping, or the "membrane" case. The energy profiles labeled "membrane" (blue curves in Fig. 2) are obtained by continuously varying the transformed polarization components ($P_y$ and $P_z$ for the 109° step, $P_x$ for the 71° switch) while allowing all other variables (the remaining polarization components, tilts and strains) to adapt to this change. We observe, as expected[12,15,35], that the tilts reverse together with the polarization. This yields the lowest-energy switching paths corresponding to the experimental case of a freestanding membrane.

Next, we consider the case of the so-called "strain clamping". In order to separate clamping effects from the effects of epitaxial misfit strain, we assume the film is thick enough so that it is fully relaxed to its rhombohedral ground state. The fully relaxed nature of the film does not, however, mean that it is free to deform; on the contrary, it is still clamped and, as dictated by the substrate, energetically favors maintaining its original state. To obtain the free energy profiles labeled "clamped" in Fig. 2a(b), e(f), we vary $P_y$ and $P_z$ for 109° switch ($P_x$ component for 71° switch) while keeping the strains $\epsilon_{11}, \epsilon_{22}$ and $\epsilon_{12}$ fixed to their equilibrium values corresponding to the initial polarization direction (before the initial 109° out-of-plane switch). All the other order parameters are allowed to relax following the polarization-switching process (Methods). One can see that strain clamping leads to slightly increased energy barriers compared with the freestanding

case, about 6% for the 109° step and 20% for the 71° step. Notably, the results obtained for the first-principles Landau potential (Fig. 2a, b) and the phenomenological model (Fig. 2e, f) are essentially equivalent with regard to the change in activation-energy barriers, highlighting the consistency of both methods. Additional free-energy calculations for the prototypical ferro-electrics PTO and BTO are presented for clamped and membrane cases (Fig S1). These calculations, consistent with previous experimental work[18], show a similar reduction in switching energy for freestanding membranes compared with clamped films, suggesting a broad applicability of the role of strain-clamping effects in ferroelectric switching.

There exist other ways in which the substrate can impact switching, namely by clamping additional order parameters (such as octahedral tilts) beyond strain. This can arise, for example, by a mismatch in rotostrictive coefficients between the film, electrode, and substrate. To quantify this effect, we introduce "strain + tilt clamping" (and varying degrees, i.e., weak vs strong, thereof) and compute the free-energy profiles while imposing additional con-straints on some order parameters in our simulations. In the limiting case of "strong strain + tilt clamping" (Fig. 2g, h) we fix all nonswitching polarization and tilt components (as well as $\epsilon_{11}, \epsilon_{22}$ and $\epsilon_{12}$) to their equilibrium values corresponding to the initial polarization direction. Moreover, we do not fully relax the switching tilt components, but interpolate them between the values corresponding to the minima of the free energy-curves (Methods). Only strains $\epsilon_{33}, \epsilon_{13}$, and $\epsilon_{23}$ are allowed to relax following the variation of the other order parameters. One can see that clamping of other degrees of freedom immediately results in a greater difference between the membrane and clamped cases (about 64% for both the 109° and 71° steps for the phase-field parameter set). We expect the true clamping effects to lie between "strain clamped" and "strong strain + tilt clamped" cases, which serve as limiting cases of clamping effects. To quantify such an

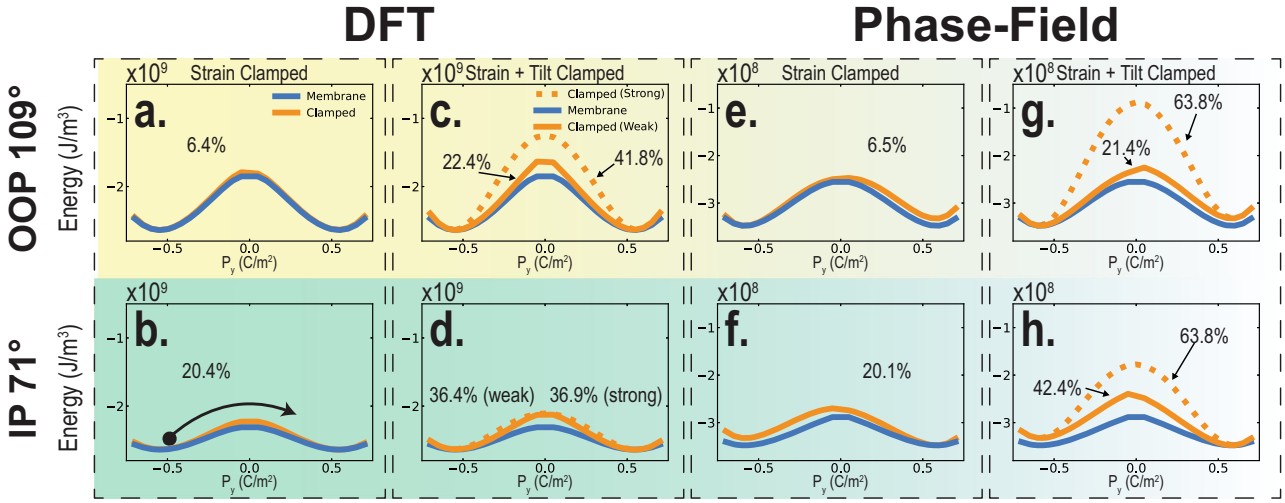

**Fig. 2 Thermodynamic calculation of switching free energy for BFO. a.** (**e.**) and **b.** (**f.**) show 109°, out-of-plane, and 71°, in-plane, switching-energy landscapes, respectively, calculated using Landau coefficients obtained from DFT (used in the phase-field model, Supporting Information Section 1) for the strain-clamped and membrane cases. **c.** (**g.**) and **d.** (**h.**) show 109° and 71° double-well potentials, respectively, calculated using the Landau potential from DFT (from the phase-field model) for strain + tilt clamped and membrane cases. To obtain the "clamped" (solid orange curves) results in panels **a.**, **b.**, **e.**, and **f.**, the in-plane strains are fixed, modeling the effect of strain clamping from the substrate. "Clamped (Weak)" (solid orange) curves in **c.** and **d.** (**g.** and **h.**) represent switching potentials derived from DFT parameters (phase-field parameters), but subject to "weak strain + tilt clamping" constraints, where all nonswitching polarization and tilt components are held fixed. "Clamped (Strong)" (dashed orange) curves in **c.** and **d.** (**g.** and **h.**) show switching potentials derived from DFT parameters (phase-field parameters), but subject to "strong strain + tilt clamping" constraints, where, all the nonswitching polarization and ferrodistortive components are held fixed and the switching components of tilts are linearly interpolated between the values corresponding to the minima of the free-energy curves. Percentages listed are reductions in maximum energy barrier for membrane vs. clamped films in each scenario. Calculations correspond experimentally to the thickest, fully relaxed, films. In the "strain clamping" case, the in-plane strains are fixed to their values in the initial state, before the 109° switch occurs. Hence, the fixed strains cannot adapt to the new polarization state after the 109° switch. The resulting state has a higher energy than the initial one, resulting in the asymmetric shape of the free-energy curves. This effect is more pronounced in the simulations using the phase-field model-parameter set (panels **e.–h.**) compared with the DFT (**a.–d.**) due to the relative magnitude of model parameters in the phase-field set being higher compared with the DFT set, which leads to the stronger-predicted clamping effect.

intermediary degree of clamping, we consider a third type of clamping termed "weak strain + tilt clamping." In this case (Fig. 2b, c, g, h), we fix the nonswitching tilt components, but allow switching tilt components to adapt freely to the change in polarization. In this case, we see a smaller reduction in the energy barrier upon releasing the film from the substrate, which more accurately describes our experimental data, discussed below.

To experimentally study the effect of clamping imposed by the substrate, we employ recent advances in chemically assisted lift-off techniques to produce freestanding BFO layers. Such techniques are rapidly emerging as an approach for tuning the lattice distortion and strain in ferroelectrics[36–42]. Several sacrificial layers have been developed, such as water-soluble $Sr_3Al_2O_6$[38], acid-solution soluble $La_{0.67}Sr_{0.33}MnO_3$ (LSMO)[39], and graphene for mechanical exfoliation[41], leading to freestanding ferroelectric films down to the monolayer limit[28], as well as integration of single-crystalline membranes[41], and flexible layers with superelasticity[42]. We demonstrate that quantitatively different features are obtained in freestanding BFO membranes vs their clamped counterparts, both in quasi-static measurements of the energetics (coercive field from hysteresis measurements) and dynamical measurements (pulsed switching studies) of the switching process.

Two types of samples were employed for this study. The first type, henceforth referred to as "clamped", is a Pt (20 nm)/SrRuO₃ (SRO 30 nm)/BFO(x nm)/SRO(30 nm) (where x ranges from 12.5 nm to 100 nm) heterostructure (Fig. 1a) epitaxially grown via pulsed-laser deposition (PLD) on SrTiO₃ (STO)[001] substrates (Methods). The second sample type, henceforth referred to as a "membrane", is a Pt/SRO/BFO/SRO/LSMO stack, that has been subsequently released from the STO[001] substrate by etching the

LSMO layer (Supporting Information Fig. S2)[39] to completely lift-off the Pt/SRO/BFO/SRO stack from the STO substrate. A supportive PDMS layer is used to then transfer the stack to a Pt/Si (001) substrate. We use an etch rate of ~1 nm/hour to dissolve the LSMO layer in order to avoid deformation or damage of the Pt/SRO/BFO/SRO heterostructure during lift-off. To verify a successful transfer, we measured via atomic force microscopy (AFM) the surface roughness of the initial film and that of the transferred freestanding membrane. Typical measurements are shown (Supporting Information Fig. S2b, c), yielding a surface roughness of 223 pm and 406 pm, respectively, indicating that the high-quality samples can be maintained during this process.

The out-of-plane (c) and in-plane (a) lattice parameters of the films both before and after lift-off were extracted from X-ray diffraction (XRD) line scans and reciprocal space maps (RSMs). The diffraction results show that the freestanding membranes still possess good epitaxial relationships between BFO and SRO layers (Supporting Information Fig. S3). The c and a lattice-parameter values of the BFO layers calculated from the XRD scans are provided as a function of BFO thickness (Fig. 3). We clearly observe that the in-plane lattice constant, a, increases and the out-of-plane lattice constant, c, decreases after the film is released from the substrate. This evolution is succinctly captured by a reduced c/a ratio compared with the epitaxial BFO films on substrates. The change in c/a for samples with BFO thickness of 8, 35, 60, and 100 nm is in the range of 1.4–1.9% (Fig. 3a), suggesting that the spontaneous distortion, and therefore polarization of BFO membranes is smaller, and the switching-energy barrier between adjacent polarization states should be correspondingly reduced[43]. Notably, the 100-nm-thick BFO-clamped film has an in-plane lattice constant (a = 3.95 Å) close to its bulk

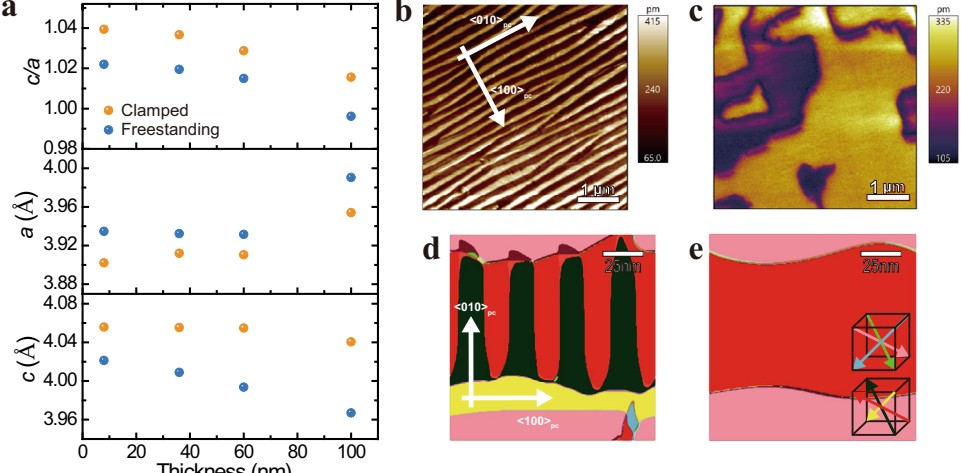

**Fig. 3 Lattice parameters and domain structure of the BFO-clamped films and freestanding membranes. a** c and a lattice parameters and their ratio c/a for the BFO films and membranes as a function of thickness. The freestanding BFO membranes exhibit decreased c, increased a, and decreased c/a ratio. **b, c,** In-plane PFM-amplitude image of 100-nm BFO film (**b**) and freestanding membrane (**c**). **d, e,** Phase-field simulation of BFO layer before (**d**) and after lift-off (**e**).

value ($a = 3.96$ Å)[44], indicating that the clamped film is nearly completely relaxed and that the effects from misfit strain ($-1.35\%$ when grown on STO)[45] are minimized at this thickness.

The impact of mechanical constraints from the substrate can be observed directly in piezoresponse force microscopy (PFM) imaging of the ferroelectric-domain structure before and after release from the substrate. In-plane and out-of-plane PFM-amplitude images for both the clamped BFO and freestanding membrane (Fig. 3b, c, Supporting Information S4a, b), reveal dramatic differences. The well-ordered 71° stripe-domain pattern of BFO in the clamped film evolves into a "blocky" 180° domain pattern with a larger domain size in the freestanding membrane. These changes are also observed in the corresponding in-plane and out-of-plane PFM-phase (Supporting Information Fig. S4c–f) images for the clamped film and membrane. Kittel's law[46] for ferroelectric domains states that the domain width scales as

$$w = \sqrt{\frac{\sigma t}{U}} \qquad (2)$$

where $\sigma, t$, and $U$ are the domain-wall energy, film thickness, and domain energy, respectively. The domain-wall energy is given by

$$U = U_{dip} + U_x + U_e \qquad (3)$$

where $U_{dip}, U_x$, and $U_e$ are the energy contributions from dipolar interactions (correlation energy), elastic energy, and depolarization energy, respectively[6]. By releasing the film from the substrate, we significantly reduce the elastic energy (correspondingly leading to an increase in the domain width), and therefore the electrostatic energy becomes the dominant energy scale. In order to minimize electrostatic energy, ferroelectric domains typically adopt configurations such that $\nabla \cdot P \approx 0$ at domain-wall boundaries, and such a condition is satisfied with 180° domains in perovskite ferroelectrics[6]. We directly observe this effect here, highlighting the dominance of electrostatic energy after removal of elastic constraints from the substrate. It is important to note that the changes in domain structure observed (Fig. 3) can be unequivocally attributed to the role of substrate clamping. In both the clamped and freestanding cases, the SRO layer is the same thickness (30 nm), so, while the SRO layer may play a small role in epitaxially constraining the BFO, its effect is present in both the clamped and freestanding cases. Therefore, any differences we observe between these two cases can be attributed to the substrate

clamping alone, and not to the SRO layer. Additional PFM images of BFO samples with thickness of 60, 35, 20, and 8 nm, before and after lift-off, are shown (Supporting Information Fig. S5). Interestingly, irrespective of the domain structure of the clamped film (i.e., either pure 71° domains or a mixture of 71° and 109° domains), all freestanding membranes feature larger domain sizes and the emergence of an exclusively 180° domain pattern, concomitant with the disappearance of the 71° and 109° domains.

Our experimental PFM results are in good agreement with the mesoscale-domain structure predicted by phase-field simulations, calculated using the same Ginzburg–Landau potential[33] as that used to calculate free-energy switching landscapes (Fig. 1) and including gradient terms to account for domain-configuration evolution (Methods). While the 100-nm-thick clamped films (Fig. 3b) exhibit two-variant stripe domains, thinner films (Supporting Information Fig. S5) exhibit four-variant domain structures. To model the domain-structure evolution, we employed phase-field simulations using both four-variant (Fig. 3d, e) and two-variant initial domain structures (Supporting Information Fig. S6). As observed in our simulations, by releasing the film from the substrate, the lateral width of the domain indeed increases dramatically, and 180° domain walls emerge in the freestanding membranes (Fig. 3e and Supporting Information S6c, d). There are two important considerations in understanding the observed domain-structure evolution: misfit strain and the clamping effect from the substrate. To disentangle the two effects, and specifically elucidate the role of clamping in the system, we turn to experimental measurements of the energetics and dynamics.

The measurements of polarization vs applied-voltage (P–V) hysteresis loops on both the clamped films and freestanding membranes were carried out at low temperature (100 K) to minimize the effects of leakage. We compare P–V hysteresis loops measured at 10 kHz for 100-nm and 35-nm BFO-clamped films (orange) and membranes (blue) (Fig. 4a and Fig. 4b). The data demonstrate that the coercive voltage, defined as the voltage at which the average polarization is zero, measurably decreases upon lift-off. The freestanding membranes also have a lower remnant polarization, consistent with the observed decrease in c/a ratio (Fig. 3a). Frequency-dependent P–V loops were measured (Supporting Information Fig S7 and Fig S8) showing less-dispersive,

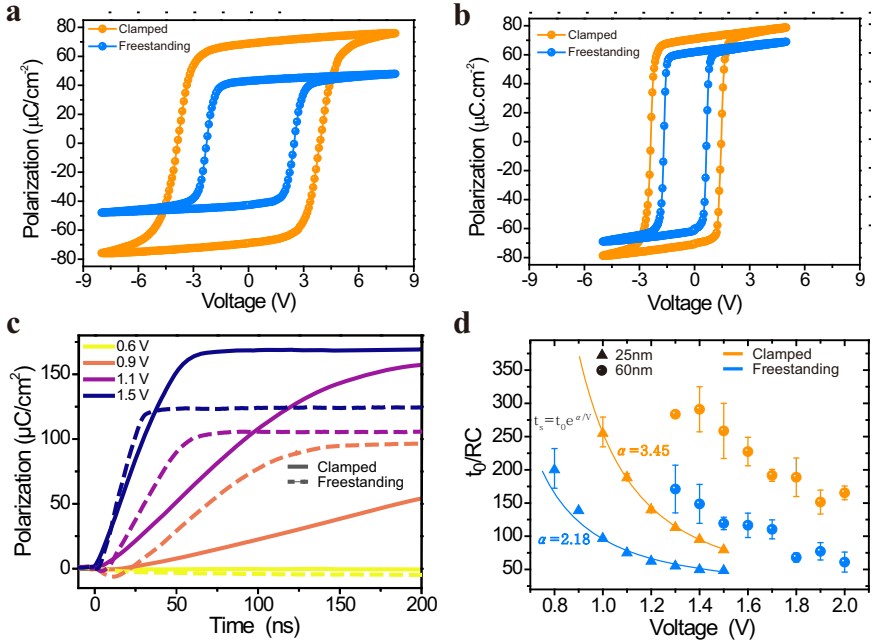

**Fig. 4 Ferroelectric switching voltage and switching dynamics of BFO films and membranes.** Ferroelectric polarization vs voltage (P–V loops) of 100 nm (**a**) and 35 nm (**b**) clamped films and membranes measured at 10 kHz. **c** Switching dynamics as a function of applied voltage for the clamped and freestanding 25 nm films. **d** Comparison of the extracted switching time (normalized by the RC-time constant) of the samples before (clamped) and after lift-off (freestanding), with BFO thicknesses of 25 nm and 60 nm. Solid lines and activation voltages (α) are shown for fits to the Merz' law for the 25 nm films. These findings show a, ~40% reduction in switching energy stemming from substrate-clamping effects.

and distinctly lower coercive voltages for the freestanding membranes when compared with the clamped films. Both 100-nm and 35-nm BFO samples show a significant decrease (~40%) in the coercive voltage after the lift-off process; indeed, this is a general feature of all the thicknesses that we studied (down to at least ~25 nm; measurements below this thickness were hampered by shorting issues and the mechanical stability of the free-standing membranes). The polarization behavior under applied voltage for both clamped and freestanding films was further investigated via PFM-based piezoelectric-hysteresis (both phase and amplitude) loops at room temperature. We can observe a distinct decrease of the switching voltage by ~40% for the samples after lift-off (Supporting Information Fig. S9), consistent with our P–V hysteresis-loop measurements at low temperature. It is important to note that, while the 100-nm-thick clamped film is known to be almost fully relaxed (Fig. 3a), reductions in switching energy persist even at this film thickness, indicating that clamping, and resistance to structural distortion *during switching*, plays a dominant role in setting the switching energetics over the effects from misfit strain. The coercive voltage is a measure of the energy required to switch the polarization, and our observed coercive-voltage ratio (~40%) between freestanding and clamped films indicates that the clamping effect lies somewhere between the limiting cases of strain clamping and strong strain + tilt clamping, and is most accurately described by weak strain + tilt clamping (Fig. 2). This is an important finding, and deserves special attention. Strain clamping alone, where the mechanical effect of the substrate only inhibits switching via the strain order parameter, is insufficient in explaining the dramatic reduction in energy observed experimentally. The substrate clamping plays an additional role, namely that the mechanical constraints imposed also inhibit variation of oxygen octahedral tilting. Only when both effects are considered, can we explain the significant reduction in switching energy observed. Furthermore, the extreme case of strong strain + tilt clamping predicts a reduction

in energy larger than that observed. This indicates that while clamping effects during switching inhibit changes in octahedral tilting, they do not completely prevent these changes from occurring.

Having established the role of clamping on switching energetics, we now turn to the dynamics of the switching process. Polarization reversal in ferroelectric thin films is known to proceed via nucleation and growth of reverse-polarized domains[21,24,47,48] and we used pulsed ferroelectric measurements[24] to directly measure the ferroelectric polarization evolution during switching. We show polarization transients for various applied voltages for a 25-nm-thick clamped film and freestanding membrane (Fig. 4c, Supporting Information Fig. S10a, b show corresponding observed ferroelectric switching displacement current). It is known that the dynamical timescale of free charge in the measurement circuit, namely RC (resistance × capacitance) time, plays a significant role in the ferroelectric switching times observed in macroscopic device structures at these timescales[24] because the RC time of the measurement circuit imposes limits on how fast one can deliver charge to facilitate polarization switching. In order to account for any such effects, we normalized all measured switching times to the measured RC-time for each device (Fig. 4d)[18,24], thereby enabling us to make meaningful comparisons across material systems (e.g., freestanding vs clamped films). Clear decreases in switching time persist even after such normalization, verifying that the observed changes are from the mechanical clamping and not changes to extrinsic-circuit parameters. The normalized switching time, defined as the time when the switched polarization reaches 90% of its saturation value, normalized by the nonswitching RC-time of the measurement circuit, was extracted for the samples with thicknesses of 25 nm and 60 nm (Fig. 4d) with capacitors of the same area. The freestanding membranes show a significant decrease in the switching time compared with the clamped films. In particular, the 60-nm-thick sample presents

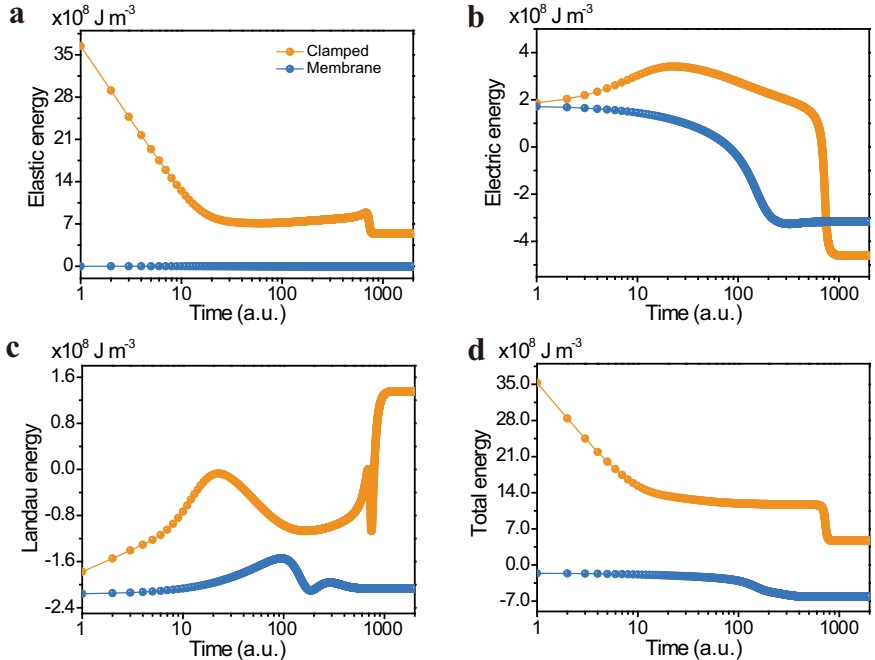

**Fig. 5 Free-energy evolution during the switching process under an externally applied voltage for clamped film and membrane cases.** Horizontal axis is the time in arbitrary unit (a.u.) and vertical axis is the average energy of the corresponding component within the whole simulation system in J m$^{-3}$. **a** Elastic energy. **b** Electrostatic energy. **c** Landau energy. **d** Total free energy that is the summation of the elastic, electrostatic, and Landau energy. For the clamped-film case, 0.4% compressive mismatch strain is considered.

an ~63% decrease in the switching time after lift-off. To quantitatively determine the role of clamping from the switching dynamics measurements, we employ Merz' law[48] to extract the ratio of the activation voltage for the clamped and freestanding films, taking care to account for RC effects in the measurement circuit. Our findings (Fig. 4d) show that the activation voltage for the 25-nm-thick clamped film is 3.45 V, while the 25-nm-thick membrane film has an activation voltage of 2.18 V, indicating that the removal of clamping effects results in a ≈37% reduction in switching energy. This finding is consistent with the considerable decrease observed in coercive voltage in the hysteresis loops (Fig. 4a, b) and energy barrier in our thermodynamic calculations (Fig. 2), showing best agreement with "weak strain + tilt" clamping scenario.

Finally, we performed time-resolved phase-field simulations (Methods[33]) to further investigate the relevant energy scales and the effects of clamping during switching. We simulated the same applied voltage (13 V) for both the clamped and membrane cases, and the evolution of elastic, electric, and Landau energy during the switching process (2000 time steps) was computed[49,50] (Fig. 5). All simulations start from an equilibrium-domain state (Methods) with polarization pointing upward. A positive voltage is then applied on top (with the bottom grounded) to switch the polarization downward. Changes in the various energy values correspond to changes in the polarization distribution within the simulation, with the most dramatic changes occurring as domains are switched. We can clearly see that the time for a freestanding membrane to switch (~200 timesteps) is significantly shorter than that for the clamped film to switch (~800 timesteps). Corroborating the true dynamic nature (opposed to quasi-static) of the experiment is the remarkable agreement between reductions in switching time predicted (~63% for the 60 nm film, and ~75% as predicted by simulation). Returning now to the fundamental aim of this paper, by examining the time-resolved evolution of the elastic, electric, and Landau energies individually (Fig. 5a–c), we use our simulations to directly interrogate the role of substrate

clamping during switching. As expected, the elastic energy of the membrane (Fig. 5a) is essentially negligible throughout the ferroelectric switching process, except for small local stresses imposed by adjacent domains. The time-resolved elastic energy of the clamped film, on the other hand, remains high, and locally peaks just before switching is completed. These results demonstrate that dynamic evolution of the polar state and accompanying structural distortions transiently modify the energy landscape. Since both nucleation and growth of reverse-polarized domains are activated processes[48] and both have exponential dependence on the activation energy, such transient changes (even moderate) in the energy landscape can have a dramatic effect on switching time. Other energy terms (Landau and electric) are also impacted by mechanical constraints, where the high elastic energy slows the evolution of the polarization, so that the higher energy state persists for a longer period of time. Finally, we address the role of oxygen octahedral tilting. The data presented (Fig. 5) include dynamical evolution of the oxygen octahedral tilts, and as such, an associated energy increase (Eq. 1). Informed by our earlier findings that the switching energy barrier lies somewhere between strain clamping and strong strain + tilt clamping (i.e., weak strain + tilt clamping), and to extract the effect on switching from the octahedral tilts, we simulate the same polarization switching (Fig. S11) without consideration of the oxygen octahedral tilts. There is a stark contrast between these two cases, with the oxygen octahedral tilts accounting for an ~10x increase in switching time over the case where oxygen octahedral tilts are removed from the simulations. This is an important finding, and the dramatic increase in switching time highlights the importance of proper consideration of all coupled order parameters in setting the dynamics and energetics of switching in BFO.

In conclusion, our work reveals the fundamental role of substrate mechanical constraints in dictating ferroelectric switching energetics and dynamics in BFO, and more broadly, for displacive ferroelectric thin-film materials in general. With the grand

challenge of achieving sub-100mV switching in ferroelectrics, clamping effects and the relative contribution to switching energetics and dynamics of all coupled order parameters must be understood. We employ a Landau free-energy formalism to conduct thermodynamic calculations modeling varying degrees of clamping effects from the substrate, both using ab initio and phenomenological models. We experimentally demonstrate a method of mitigating clamping effects by lifting-off SRO/BFO/SRO trilayers from an STO substrate. Other methods, for example, by tuning device aspect ratio[51], may provide additional pathways to mitigate clamping. Here, we observe a marked evolution of crystal and domain structure, consistent with changes in elastic constraints, and show that the energetics and dynamics of the system drastically change after lift-off. We observe a significant reduction in switching voltage and improved switching speeds for freestanding membranes relative to clamped films. The origins of the changes observed are better understood with the help of phase-field simulations, where the dynamic elastic energy and oxygen octahedral tilts play a predominant role in slowing polarization reversal.

## Methods

**Thermodynamic calculations.** The calculations of free-energy profiles are performed using in-house code for solving the system of equations $\frac{\partial \varphi_i}{\partial t} = -L_\varphi \frac{\partial f}{\partial \varphi_i}$[52], where $f$ is defined by Eq. 1 (main text), $\varphi_i$ is the order parameter, and $L_\varphi$ is the kinetic coefficient describing the rate at which $\varphi$ approaches its equilibrium value. Since, in these simulations, we are only interested in the equilibrium values of the order parameters and not the trajectory by which it is reached, the values of $L_\varphi$ are set to 1 in the reduced units (we checked that the choice of $L_\varphi$ does not affect the resulting equilibrium state of the system).

We perform the calculations of the free-energy profiles for the polarization-switching path in which 109º out-of-plane switch ($P_y$ and $P_z$ polarization components are reversed) is followed by the 71º in-plane switch ($P_x$ component is reversed). This path is chosen based on PFM experiments on BFO/SRO heterostructures previously reported[14], in which the initial polarization-switching event follows the aforementioned step sequence. In the following, we will only discuss the 109º out-of-plane switch, though the same considerations can be applied to 71º in-plane switch.

For the freestanding BiFeO$_3$ membrane, we assume that during the polarization-switching process, all order parameters can freely evolve and adapt to the instantaneous values of the switching polarization components. For example, for 109º out-of-plane switch, at each value of $P_y = -P_z$ between $-0.7$ and $0.7$ C/m$^2$, we allow $P_x$ as well as all the components of $\theta$ and the strain tensor, to evolve to their preferred values as dictated by $\frac{\partial \varphi_i}{\partial t} = -L_\varphi \frac{\partial f}{\partial \varphi_i}$. Then, we use the relaxed values of P, θ and strain to compute the energy of the system, at each step of the switch. For completeness, the evolution of all order parameters with varied $P_y$ is shown in Fig. S12 of the Supporting Information (for a phase-field parameter set).

**"Strain Clamped" vs "Strain + Tilt Clamped".** In the simulations of the films clamped by the substrate, we consider three possible clamping effects. First, we assume that the substrate clamps only $\epsilon_{11}, \epsilon_{22}$ and $\epsilon_{12}$ components of the strain tensor ("strain clamping" case). To reflect this in our simulations, we fix $\epsilon_{11}, \epsilon_{22}$ and $\epsilon_{12}$ to their equilibrium values corresponding to the initial direction of the polarization and to the considered set of model parameters (either DFT or phase field). Then, similarly to the case of the freestanding membrane, for 109º out-of-plane switch, we vary $P_y = -P_z$ components of the polarization between –0.7 and 0.7 C/m$^2$ and for each $P_y$ value, we optimize $P_x, \theta_x, \theta_y, \theta_z$ as well as the unclamped components of the strain tensor. The evolution of the order parameters with varying $P_y$ is presented in Fig. S12 of the Supporting Information together with that of freestanding case.

Next, we consider the possibility that the presence of the substrate can have an additional clamping effect on other order parameters, such as FeO$_6$ octahedral tilts ("strain + tilt clamping"). In particular, we study two cases, to which we refer in the following as "weak strain+tilt clamping" and "strong strain + tilt clamping". In the case of weak strain + tilt clamping, in addition to $\epsilon_{11}, \epsilon_{22}$ and $\epsilon_{12}$, we fix the nonswitching components of polarization and tilts ($P_x$ and $\theta_x$ for 109º out-of-plane switch) to their initial equilibrium values. Then, we vary $P_y = -P_z$ components and for each considered $P_y$ value, we optimize $\theta_y, \theta_z$ as well as the remaining components of the strain tensor ($\epsilon_{33}, \epsilon_{13}$ and $\epsilon_{23}$). The evolution of all order parameters with Py is presented in Fig. S13 of the Supporting Information. To simulate the case of strong strain+tilt clamping, we first identify the equilibrium values of all order parameters for both sets of the Landau potential coefficients (DFT and phase field). Then, we vary $P_y = -P_z$ and $\theta_y = -\theta_z$ between their equilibrium values corresponding to up and down polarization directions

while we keep $P_x, \theta_x$, as well as $\epsilon_{11}, \epsilon_{22}$ and $\epsilon_{12}$ fixed to their initial values. For each value of $P_y$ and $\theta_y$ we relax the unclamped components of the strain tensor ($\epsilon_{33}, \epsilon_{13}$ and $\epsilon_{23}$). The evolution of the order parameters is shown in Fig. S14 of the Supporting Information.

**Film growth.** The oxide heterostructures BFO/SRO/LSMO or SRO/BFO/SRO/LSMO were grown on single-crystalline (001) STO substrate by pulsed-laser deposition at 650–720 °C with focused laser fluence ~1.2 J cm$^{-2}$ in 100–160 mTorr oxygen pressure and cooled down to room temperature in 400 Torr oxygen pressure. To protect the film during the lift-off process, a 20-nm Pt layer was deposited on top of the SRO or BFO layer by magnetron sputtering. The top SRO layer was patterned into a circular top electrode and the bottom SRO layer served as a bottom electrode for ferroelectric switching testing.

**Lift-off process.** PDMS stamps were cut into 8 mm × 8 mm × 1.5 mm from a commercial specimen (Gelfilm from Gelpal). Then they were stacked tightly onto the film. After floating the PDMS/films in an etching solution (low-concentration HCl solution (0.3 vol%) mixed with 0.1 mol mL$^{-1}$ potassium iodide) for several hours, the LSMO dissolved to lift the freestanding film off STO substrate, which was washed with deionized water and dried with N$_2$ gas. The samples were then moved onto Si/Pt substrate. The entire stack was annealed at 110 °C for 30 min to promote adhesion at the film/new substrate interface. After cooling to 70 °C and peeling off the PDMS stamp with tweezers, the transferred membrane on Si/Pt substrate was obtained.

**X-ray diffraction (XRD) and reciprocal space mapping (RSM).** The films before and after lift-off were measured with a Panalytical Empyrean diffractometer (Cu-Kα$_1$, 1.540598 Å), using a hybrid, two-bounce primary monochromator on the incident beam. RSM of the samples was acquired with the same incident beam optics and a PIXcel$^{3D}$ position-sensitive detector, using the frame-based 1D mode with a step time of 10 s.

**Piezoresponse force microscopy (PFM).** PFM was performed with an atomic force microscope (Asylum Research Cypher, Santa Barbara, CA), conductive AFM probe (Nanoandmore, DT-NCHR, Watsonville, CA) with the DART mode. The typical contact-resonance frequency is 260 kHz and its higher harmonics.

**Polarization testing.** For polarization-switching measurements (P–V loops), we used the patterned circular top electrodes with diameter of 16 μm at the frequency of 1 Hz–100 kHz at both room temperature and 100 K with Precision Multiferroic tester (Radiant Technologies).

**Phase-field simulation.** The current phase-field model for ferroelectric freestanding film is an extension to our previous model for bulk and epitaxial thin-film simulations[32,33,50,53,54] in which we use the spontaneous polarization $\mathbf{p} = (p_1, p_2, p_3)$ and oxygen octahedral tilt $\mathbf{\theta} = (\theta_1, \theta_2, \theta_3)$ as the order parameters. A temporal evolution of the order parameters can be obtained by solving the time-dependent Ginzburg–Landau equation[55,56]:

$$\frac{\partial p_i}{\partial t} = -L_p \frac{\delta F_{tot}}{\delta p_i}, (i = 1, 2, 3) \quad (4)$$

$$\frac{\partial \theta_i}{\partial t} = -L_\theta \frac{\delta F_{tot}}{\delta \theta_i}, (i = 1, 2, 3) \quad (5)$$

in which $L_P$ and $L_\theta$ are the kinetic coefficients, $t$ is time, $F_{tot}$ is the total free energy of BFO membrane. Owing to the lack of experimental data to which to fit $L_p$ and $L_\theta$, we set both $L_p$ and $L_\theta$ to be 1 after normalizing all coefficients to unitless values. The expression for the total free energy is given as:

$F_{tot} = \iiint_{V_{BFO}} (f_{land} + f_{grad} + f_{elec} + f_{elast}) dV$. All equations with repeating subscripts follow the Einstein summation notation, and the comma in subscript means spatial differentiation, e.g., $p_{i,j} = \frac{\partial p_i}{\partial x_j}$. $f_{Land} = \alpha_{ij} p_i p_j + \alpha_{ijkl} p_i p_j p_k p_l + \beta_{ij} \theta_i \theta_j + \beta_{ijkl} \theta_i \theta_j \theta_k \theta_l + t_{ijkl} p_i p_j \theta_k \theta_l$ represents the local free-energy density, which includes the Landau energy for polarization, oxygen octahedral tilt, and the coupling terms, $f_{grad} = \frac{1}{2} g_{ijkl} p_{i,j} p_{k,l} + \frac{1}{2} \kappa_{ijkl} \theta_{i,j} \theta_{k,l}$ represent the gradient energy, which includes the gradient energy for both polarization and oxygen octahedral tilt, $f_{elec} = -\frac{\varepsilon_0 \varepsilon_{ij}^b}{2} E_i E_j - E_i p_i$ is the electrostatic energy, where $\varepsilon_{ij}^b$ is the background dielectric constant[57–60], $E_i$ is the electric field obtained by solving the electrostatic equilibrium equation $\varepsilon_0 (\varepsilon_{11}^b \frac{\partial^2 \Phi}{\partial x^2} + \varepsilon_{22}^b \frac{\partial^2 \Phi}{\partial y^2} + \varepsilon_{33}^b \frac{\partial^2 \Phi}{\partial z^2}) = \frac{\partial p_1}{\partial x} + \frac{\partial p_2}{\partial y} + \frac{\partial p_3}{\partial z}$, $\Phi$ is the electrical potential, and $f_{elast} = \frac{1}{2} C_{ijkl} (\varepsilon_{ij} - \varepsilon_{ij}^0)(\varepsilon_{kl} - \varepsilon_{kl}^0)$ is the elastic energy, in which $\varepsilon_{ij}$ is the total strain distribution obtained by solving the mechanical equilibrium equation $\sigma_{ij,j} = 0$, $\sigma_{ij} = C_{ijkl}(\varepsilon_{kl} - \varepsilon_{kl}^0)$, and the eigenstrain is connected to the order parameters $\varepsilon_{ij}^0 = \lambda_{ijkl} \theta_k \theta_l + Q_{ijkl} p_k p_l$. There are two differences between our freestanding film and our epitaxial thin-film model[33], both of which pertain to solving for the mechanical equilibrium state. First, the clamped thin film

has a fixed displacement bottom and traction-free top surfaces, while for the freestanding film, both the top and bottom surfaces are set as traction-free boundaries, as shown in Eq. (6) owing to the nature of the freestanding membrane

$$\begin{cases} \sigma_{i3}|_{z=0} = 0 \\ \sigma_{i3}|_{z=h_f} = 0 \end{cases}, \; (i = 1, 2, 3) \qquad (6)$$

where $h_f$ means the membrane thickness. Second, the in-plane macroscopic strain in clamped thin film is controlled by the misfit of the substrate, while for the freestanding film, the in-plane macroscopic strain is set to be the average eigenstrain calculated from the current order-parameter distribution, since overall the free-standing film is in a stress-free state.

We start our simulation from random noise for both the polarization and oxygen octahedral tilt in a 128 nm × 128 nm × 30 nm (each simulation grid $\triangle x = 1$nm) thick clamped thin film with 0.4% compressive biaxial misfit and 0 applied electric field. The system is relaxed to an equilibrium state as shown in Fig. 3d. Then the same domain structure is used as an input to the freestanding film simulation without applied electric field, from which we get the domain structure of Fig. 3e. Next, starting from this equilibrium state, we first apply an instantaneous negative electric potential, −10V, on both films' top surface (bottom-grounded) to pole the system fully upward, then remove the potential to relax the system back to equilibrium. The domain state at this stage is 71 stripe domains in the clamped film and single domain in the free-standing film. Finally, an instantaneous 13 V electric potential is applied on both films' top surface (bottom still grounded) to switch the domains downward, during which process we keep track of the energy evolution as shown in Fig. 5.

The film thickness is 30 nm, or 30 layers of grid points, for both the clamped and freestanding film. In the clamped-film case, there are 6 layers of grid points of air layer above and 10 layers of grid points of substrate layer below the film along the z direction. While in the freestanding film case, there are 6 layers of grid points of air layer above and 0 layer of grid point of substrate layer below the film. More details regarding our numerical simulations and all physical parameters in the above equations are listed in the Supporting Information.

**Reporting summary**. Further information on research design is available in the Nature Research Reporting Summary linked to this article.

## Data availability
The data that support the findings of this study are available from the corresponding authors upon reasonable request.

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

## Acknowledgements

The work at Berkeley is supported by ASCENT, one of the six SRC-JUMP centers. Support from Intel Corporation under the FEINMAN Program (E.P.) is also gratefully acknowledged. A.F. acknowledges support from the Army Research Office under Grant W911NF-21-1-0118. Q.W.S. acknowledges support from International Visiting Program for Excellent Young Scholars of SCU and the National Natural Science Foundation of China (No. U20A20212). D.P. acknowledges support from the European Union's Horizon 2020 research and innovation program under the Marie Skłodowska-Curie grant agreement No. 79712 and from the National Science Foundation under grant Grant DMR-1708615 for work done at Berkeley. L.W.M. and R.R. acknowledge support from the Army Research Office under the ETHOS MURI via cooperative agreement W911NF-21-2-0162. N.F and J.Í. acknowledge support from the Semiconductor Research Corporation and Intel, via contract no. 2018-IN-2865. The thermodynamic calculations and phase-field simulations of X.C. and L.-Q.C. are supported by the U.S. Department of Energy, Office of Science, Basic Energy Sciences, under award no. DE-SC0020145. Y.L.H acknowledges the financial support from "Center for the Semiconductor Technology Research" from The Featured Areas Research Center Program within the framework of the Higher Education Sprout Project by MOE in Taiwan, and the Ministry of Science and Technology, Taiwan, under Grant MOST 110-2634-F-009-027. R.-C. P. acknowledges supports from the Natural Science Foundation of China (grant no. 51902247) and Natural Science Foundation of Shanxi Province (grant no. 2020JQ-059).

## Author contributions

E.P., Q.W.S., Y.L.H., and R.R. designed the experiments. Q.W.S., H.R.Z., X.X.H., and S.D. carried out the synthesis; Q.W.S., X.Chang, and D.P. carried out the lift off studies; X.C. and R.C.P. carried out the phase field computation while N.F. and J.I. carried out ab initio computation studies. Q.W.S., E.P., A.F., and A.Q. carried out the ferroelectric and switching dynamics studies as well as the low temperature measurements; Q.W.S. and Y.L.H. wrote the manuscript. E.P., L.W.M., L.Q.C., J.I., D.N., I.Y., and R.R. wrote majority portions of the manuscript and carried out critical internal reviews of all the versions of the manuscript.

## Competing interests

The authors declare no competing interests.
