## [Peer review file · Nature Communications]

REVIEWER COMMENTS

Reviewer #1 (Remarks to the Author):

In this manuscript, Shi et al. reports the ferroelectric switching kinetics in clamped and freestanding BiFeO₃ films. They conducted various experiments, e.g. PFM, X-ray diffraction, and phase-field simulations, to characterize the switching behaviours. They conclude that the morphology, domain switching behaviours, switching time, coercive voltage, switching voltage, are all different between clamped/freestanding films.

While the idea of this research is important to the field, it is not completely new, considering similar researches have been reported before (Phys. Rev. Lett. 118, 017601 (2017), Science Advances, eabc7156 (2020)). While the various switching behaviours, switching time, switching voltages, switching pathway free energy have been reported to be altered by the clamp/unclamp situations, the new findings from the current manuscript are overclaimed. Considering this, I am not able to recommend the publication of this work in Nature Communications. However, I will reserve this decision upon the authors can fully justify the novelty of the manuscript and the differences between this work and previously published works.

While this work has been presented in a good way, considering the characterizations, switching dynamics, and simulations are well aligned with the conclusions, some issues need to be addressed:

1. Domain size is severely altered due to the unclamped feature of the freestanding films from fig 3. It may be attributed to the release of the substrate constrain. However, the film is not 100% relax while a thin layer of SRO is still presented. In this case, it requires evidence to confirm the negligible effect of the SRO layer on the BFO film.
2. Although the switching kinetic has been simulated by phase field, the detailed switching procedure regarding the behaviours of 71/109 domains in the clamped film and 180 domains in the freestanding films needs to be discussed. These switching behaviours should also be linked to the observed change of ferroelectric switching voltage and dynamics.

Reviewer #2 (Remarks to the Author):

This paper is dedicated to a deliberate experiment to understand the role of oxygen octahedral in switching process of BiFeO₃ ferroelectric domain. In the experiment the author gains ~60% switching speed by controlling the oxygen tilting. Moreover, the significance of the octahedral tilting is demonstrated elegantly by the phase field simulation. It provides us a brand new point of view to dig into the basic mechanism of the domain switching dynamics. It is an interesting work and worthy of publication in Nat. Commun, and I recommend that the paper can be accepted for publication, subject to answering the following questions.

Q1: When people talk about lattice dynamics, they always mention phonon behaviors in crystal, which are momentum and frequency dependent. The author mentioned this as well in the introduction, however they did not discuss it anymore in the following part of the paper. It would be great if the author could try to discuss the phonon behavior in the case of "tilting clamping" of oxygen octahedral. Otherwise, the title of lattice dynamics is somehow misleading.

Q2: The details of the "tilting clamping" are not demonstrated in the methods part. Would the author add these details?

Q3: The coercive voltage is determined by many effect; it is not convincing to identify the experimental clamping effect as it lies somewhere between strain clamping and strain+tilt clamping

from the energy point of view, although it seems straight forward.

Q4: The calculated free energy switching pathway for BFO is neither the same as the calculation in Physical Review B 98,085107 (2018) nor Nature 516, 370 (2015). Would the author explain the difference?

Q5: The significance of octahedral tilting in the ferroelectric switching behavior of BFO was demonstrated in previous papers, for example L.Bellaiche et. al. Physical Review Letters 99,227602(2007), L.Q.Chen, et. al. Physical Review B 98,085107 (2018) and J. Hlinka, et. al. Physical Review B 96, 174110 (2017). The timescale of experiment in Fig. 4 is in nano-second, while it is in pico-seconds in Fig. 5. Although the discrepancy of the timescale is possibly stems from the limitation of grid size in phase field simulation, it brings forth the concern that the experiment is quasi-static process comparing to the phase field simulation. In which case, the contribution of octahedral tilting is well known from static calculations.

Reviewer #3 (Remarks to the Author):

The manuscript under consideration deals with experimental and numerical investigations of ferroelectric switching in thin films made of bismuth ferrite (BiFeO₃, BFO). Goal is to reduce the energy barriers required to electrically switch domain patterns of the films as well as to reduce the associated switching times. Key idea is to relieve the BFO film from its substrate and therewith to allow for energetically more favorable switching paths. The authors analyze the switching of both clamped films (with substrate) and freestanding films (without substrate) experimentally and numerically. On the experimental side, piezoresponse force microscopy, X-ray-diffraction and pulsed ferroelectric measurements are employed to study in detail the microstructure and ferroelectric response of the films (e.g., lattice parameters, ferroelectric domain structures, hysteresis). On the numerical side, phase-field modeling is used to study the evolution of microstructures under applied electric fields (ab-initio calculations are also presented, but only briefly addressed). Based on experiments and numerical simulations, the authors quantify the substrate's impact on coercive fields, saturation polarization and associated switching times. It is shown that freestanding films could significantly reduce the necessary energy barriers and the times related to ferroelectric switching. Furthermore, by numerically comparing classical clamping (termed "strain clamping" in the manuscript) and additional freezing of the oxygen octahedral tilts, it is found that tilting has a large impact on the ferroelectric switching of the films.

The authors document a number of very interesting and novel findings on a theoretically and practically relevant subject. By combination of detailed experimental and numerical studies, they have been able to study in-depth various complex phenomena in BFO films that, according to my knowledge, haven't been analyzed/demonstrated to that extent before. The methods used are at the state of the art and mostly well documented (see below for additional comments/questions in that regard) and the conclusions are sound.

Below, I list a number of remarks and questions of which I hope that they will prove useful for the authors to further improve the quality of their manuscript.

1) While certainly clear to most of the readers, the authors may want to introduce BiFeO₃/BFO as bismuth ferrite in the abstract/introduction.

2) In the abstract, the statement "[...] it is found that removing the constraints imposed by the requirement of macroscopic continuity of deformation (also known as the von Mises criterion in the deformation of solids [...])" may be misleading. In fact, "removing the constraints imposed by the requirement of macroscopic continuity of deformation" could be understood as if the authors were taking into account macroscopic discontinuities, such as cracks. However, the two scenarios

considered ("freestanding" and "clamped") still constitute continuous configurations. Also the notion of the "von Mises criterion in the deformation of solids" could be further explained (or removed) as it doesn't seem to be a classical criterion employed in solid mechanics for the present purposes.

3) A useful and interesting outcome on the side of numerical modeling is certainly the significance of the oxygen octahedral tilt as an additional order parameter when modeling BFO. Since the established and widely used Ginzburg-Landau theories for ferroelectrics (including BFO) often employ as an order parameter the polarization only, the authors may want to elaborate a bit more on the ferrodistoritive effect; perhaps by adding some more explanatory information to the supplementary material of the manuscript. In that context, it would also be appreciated if the authors could motivate the used model in a more detailed way. From what I understand, the tilts represent rotations. What kind of reference frame are the angles related to? How do the authors treat the periodicity of rotations w.r.t. 2π and its impact on gradient energy and interpolation? How do the authors motivate the corresponding equation for the ferrodistoritive strain? How about the emergence of couple stresses? Furthermore, please provide a literature reference related to the model written down in Equation 1.

4) When describing the characteristic switching process of BFO, the authors could add more clarity by stating what plane they are referring to (i.e., to what entity "out-of-plane" and "in-plane" are related to).

5) The sentence "We first calculate the energy associated with "strain-clamping" (Fig. 2a-d), i.e. to obtain the curves labeled "clamped", we impose that strains ϵ_{11} , ϵ_{22} and ϵ_{12} be fixed at the values corresponding to the initial state" should be reformulated. Same applies to "We obtain similar switching paths the membrane case, [...]". The latter statements shall also conclude with "71° step". Later on the same page: "With regards the change".

6) Figure 1 (in particular 1c): As the authors later refer to the Cartesian coordinates of the polarization vector, a coordinate system shall be added already here.

7) Figure 2: (i) A comment on why we observe unsymmetric landscapes for most of the clamped cases would be appreciated. (ii) In e and f, we don't seem to have reached an energy minimum yet. (iii) Caption: "This results in higher energy barriers, observed".

8) The authors state that they are "using now a complete Ginzburg-Landau potential resolved via the Landau-Khalatnikov dynamical equation [46]". What precisely does that mean? First, the cited paper proposes a multiferroic model, including magnetic and magnetoelectric effects. Second, why is the thus far used model not sufficient anymore?

9) Figure 3: (i) Could the authors provide a color scale for the shown plots? (ii) What are the spatial dimensions of the numerically simulated regions? (iii) What is the thickness of the film considered in the numerical simulations? (iv) The authors have employed a periodic representation the film by using periodic boundary conditions. This shall be stated explicitly in the manuscript. Furthermore, did the authors perform a study w.r.t. the size of the periodic region? (v) The orientations of the stripe domains is different in (b) and (d): Could the authors explain? (vi) Caption: "(a)" and "(b)" shall be replaced with "(b)" and "(c)".

10) The statement "The saturation polarization of the freestanding membrane is lower than that of the clamped film, consistent with our observed reduction in c/a ratio (Fig. 3) and quasi-static hysteresis loops (Fig 4a,b)." is more or less a repetition of what has just been noted a few lines earlier.

11) The abbreviation "RC" may be introduced.

12) "In order to account for any such effects, we normalized all switching times to the measured RC-time constant for each device (Fig. 3d)": The number of the figure shall be replaced with "4".

13) Figure 4 (caption): "free-standing films 25nm samples".

14) When introducing the model used for the numerical simulations, the authors refer to "(Methods, [49])". The cited models seem to be different, however. What kind of model has been used? Wouldn't it be more convenient so select only one model and to deduct any simulations based on that (see also remark 8).

15) The way in which parts of the numerical simulations were performed could be described in a bit more detailed way. For example, when saying "All simulations start from an equilibrium domain state with polarization pointing downwards, a positive voltage is then applied on top to switch the polarization upwards": What is the potential at the bottom? How was that potential at the top applied (ramped, instantaneous)? What is the influence of the chosen boundary conditions on the dynamics of the different domain structures? What are the initial conditions for the tilts?

16) The style of Fig. 11 could be adjusted to the style of Fig. 5 as both show similar results.

17) In the conclusion: "all coupled order parameters much be understood".

18) In the methods section, under "phase-field simulation": (i) The " θ " indicating the tilt is a vector and shall thus be printed in boldface. (ii) "that is couple to the elastic" (iii) "of the order parameter spatial distribution". (iv) In the first two equations: In the partial derivative of the polarization w.r.t. time, the " t " should be on the same level as the partial sign. (v) In the first two equations: The Ginzburg-Landau theory is local in the sense that " f " should appear on the right-hand side rather than " F_{tot} ". (vi) In the first two equations: The authors are using the same mobility parameter " L " in both equations. Is that true? How are these parameters adjusted, in particular for the ferrodistorptive part? (vii) " L is the kinetic coefficient related to the domain wall mobility": What domain wall mobility? (viii) " F_{tot} " does not contain any dielectric energy. Why? I would expect that it is of major relevance here. (ix) "landan". (x) " f_{grad} " misses " $1/2$ ". (xi) " f_{elect} " should read " f_{elec} " (as before). (xii) " f_{elastic} " should read " f_{elast} " (as before). (xiii) The sentence "There are two differences between the current free-standing film and our previous epitaxial thin film model [32], all of which are in the elastic part" could be misleading as the "elastic part" has just been associated with the energy. (xiv) The numbering of Equation "2" may be adjusted. (xv) In Equation 2: " $z=0$ " instead of " $z=1$ "?

19) Supporting Information, Section 1: While formulas are stated in full index notation, the parameters listed in the table are given in Voigt notation. Please provide a means to relate them. Furthermore, please provide associated references.

20) Supporting Information, Section 2: It is not clear to me how the authors are treating the mechanical boundary conditions. Could they describe that in a more detailed way? In particular, I wonder how they realized the in-plane stress-free conditions related to the freestanding films during the whole process of the simulations.

21) Follow-up comment: The results reported in the paper heavily rely on numerical simulations. To allow for better reproducibility, I would like to ask the authors to provide more background information on the used numerical schemes (theory, algorithms, discretizations, boundary conditions, ...) in the supplementary material of the manuscript.

**Response to Referees Letter for the manuscript "The role of lattice**
**dynamics in ferroelectric switching"**

Dear Editor and Referees,

Thank you very much for your recent comments concerning our manuscript
(NCOMMS-21-31371A) entitled "The role of lattice dynamics in ferroelectric
switching". Based on the comments and suggestions received, we have made
careful modifications on the original manuscript, point by point. All the
amendments in the manuscript have been highlighted in Yellow. The
responses to the reviewers' comments/questions are presented as follows:

**Summary of changes:**

Following insightful comments from all three reviewers, we have made
substantial edits to the manuscript to address their questions and improve the
paper. We provide a brief summary of changes here, and include an
highlighted version of the text indicating where said changes have occurred.
Following this summary, we provide detailed, enumerated answers to each of
the reviewer's questions.

- 1. We have updated the manuscript to stress the novelty of our findings. In
particular, we emphasize the importance of understanding the impact of
substrate clamping on ferrodistorive order in the BiFeO₃ system, and
how this so-called "strain + tilt clamping" impacts ferroelectric switching.
We go far beyond previous works by noting that strain clamping alone is
insufficient in explaining our experimentally observed data. We more
clearly introduce the notion of strain + tilt clamping and have added
additional theoretical analysis to study varying degrees (weak vs strong)
of said clamping. We have also established a more robust connection
between the experimentally observed data and that predicted by theory.
a. These changes highlighted by a new Figure 2.
b. Throughout the manuscript there is updated discussion of strain
+ tilt clamping, and the context it provides for interpreting our
experimental data.
2. We have updated Figures 1, 2, and 3 to make more clear our findings
and address a number of reviewer comments.

We have substantially improved our discussion of the theoretical modeling,
adding updated text, Methods and Supporting Information Section 1, and 3

and Figures S12 – S15.

**Reviewer #1** (Remarks to the Author):

In this manuscript, Shi et al. reports the ferroelectric switching kinetics in
clamped and freestanding BiFeO₃ films. They conducted various experiments,
e.g. PFM, X-ray diffraction, and phase-field simulations, to characterize the
switching behaviours. They conclude that the morphology, domain switching
behaviours, switching time, coercive voltage, switching voltage, are all different
between clamped/freestanding films.

While the idea of this research is important to the field, it is not completely new,
considering similar researches have been reported before (Phys. Rev. Lett.
118, 017601 (2017), Science Advances, eabc7156 (2020)). While the various
switching behaviours, switching time, switching voltages, switching pathway
free energy have been reported to be altered by the clamp/unclamp situations,
the new findings from the current manuscript are overclaimed. Considering this,
I am not able to recommend the publication of this work in Nature
Communications. **However, I will reserve this decision upon the authors
can fully justify the novelty of the manuscript and the differences
between this work and previously published works.**

We thank the referee for the comments. The two papers that the referee has
cited, which consider ferroelastic switching pathways and energetics in
constrained and unconstrained relaxor ferroelectrics, are now cited in the
manuscript. In some sense, this is complimentary to the work we have
presented in our paper, which discusses the role of clamping in thin-films
ferroelectrics, particularly BiFeO₃. Our paper focuses on the microscopic
details of how the octahedral tilts and substrate clamping can impact the
energetics and particularly the dynamics of ferroelectric switching. Importantly,
our manuscript details how mechanical constraints from the substrate
influence ferroelectric switching through three coupled order parameters:
elastic, ferroelectric, and ferrodistorive (oxygen octahedral tilting). We present
an “energy breakdown” (Figure 2, Figure 5, Supplementary Figure S11),
outlining how these three order parameters combine to dictate the energetics
of switching. Furthermore, we present dynamic studies, both experimental (Fig
4) and theoretical (Fig 5), which demonstrate how there is a dynamic
mechanical clamping process which modifies the energy landscape in
time-domain *during switching*. These findings provide novel insight into the
time-resolved nature of clamping effects and into the impact of the substrate
on the three coupled order parameters in the BiFeO₃ system.

Additionally, we have updated the manuscript to emphasize a key finding,
namely, the importance of oxygen octahedral tilts and the role of “strain + tilt
clamping” in influencing ferroelectric switching. It is impossible to exaggerate
the importance of the octahedral tilts in BFO, particularly when it comes to
discussing ferroelectric switching and its impact in the magnetic structure of
the material. It is well established (Phys. Rev. B **71**, 060401(R) (2005)) that the
direction of the magnetization in BFO is determined by the axis of the
octahedral tilts, the magnetization sign being controlled by the sense of
rotation of the O₆ groups. Thus, deterministic control of the magnetization
requires deterministic control of the tilts, so it is critical to understand how
polarization switching affects the tilting pattern and vice versa. As shown in the
literature (*Nature* **516**, 370–373 (2014)) and further evidenced in the present
work, the tilts (which constitute a primary order parameter of BFO, as robust as
the polarization itself and strongly coupled to it) play an active role in
determining the ferroelectric switching path, which in turn determines the
evolution of the magnetization. Previous work (L.Bellaiche et. al. Physical
Review Letters 99,227602 (2007), L.Q.Chen, et. al. Physical Review B
98,085107 (2018) and J. Hlinka, et. al. Physical Review B 96, 174110 (2017),
now cited in the manuscript) has established how oxygen octahedral tilts
influence switching pathways and ground state energies in BFO, and, as the
referee points out, clamping effects are known to play an important role in
polarization switching. Our work fills an important void, detailing the nuanced
interplay between these three order parameters in conjunction with the
substrate clamping effect. Moreover, through proper consideration of all three
coupled order parameters, we show that, in fact, strain clamping alone is
insufficient in explaining the reduction in energy and switching time observed
in experiment. Only when we consider “strain + tilt clamping” (and varying
degrees thereof) are we able to establish the role of mechanical constraints
from the substrate in setting the energetics and dynamics of ferroelectric
switching in BFO.

Regarding changes to the manuscript, we have updated the introduction and
abstract in order to stress the key findings of our work.

While this work has been presented in a good way, considering the
characterizations, switching dynamics, and simulations are well aligned with
the conclusions, some issues need to be addressed:

1. Domain size is severely altered due to the unclamped feature of the
freestanding films from fig 3. It may be attributed to the release of the substrate
constrain. However, the film is not 100% relax while a thin layer of SRO is still

presented. In this case, it requires evidence to confirm the negligible effect of
the SRO layer on the BFO film.

The referee raises an interesting point. We would like to address it in detail
here. There are two types of constraints that we envision: one that is from
epitaxial constraints which is microscopic (in that epitaxial strain relaxes in
~ 10 s nm), and the other from the substrate clamping which is macroscopic
(and persists at all film thicknesses of relevance to our work). Clamping is
inversely proportional to the thickness of the layers (substrate $\sim 500\mu\text{m}$, SRO
30nm, BFO 30-100nm) as well as the elastic modulus of the individual layers
(*Nature Mater* **2**, 43–47 (2003)). Since the SRO and BFO layers are
approximately the same thickness, as well as approximately the same
modulus, we do not expect the SRO layer to impose a strong clamping effect,
unlike the substrate, which is 500 microns thick. Further, it is important to note
that the findings in **Fig 3** can be unequivocally attributed to the role of
substrate clamping. In both the clamped and freestanding cases, the SRO
layer is the same thickness (30nm), so, while the SRO layer may play a small
role in epitaxially constraining the BFO, its effect is present in both the clamped
and freestanding cases. Therefore, any differences we observe between these
two cases can be attributed to the substrate clamping alone, and not to the
SRO layer. We have updated the manuscript in the following ways:

- 1. Updated Fig 1 to highlight “substrate clamping” in lieu of simply
“clamping”, and removed the SRO layer from the red highlighted region
notating clamping.

2. We have made clearer our discussion to emphasize the fact that, while
the SRO layers do may play a role in epitaxially constraining the BFO,
mechanical clamping from the substrate is the sole cause of the
changes in domain structure observed in Fig 3.

*“It is important to note that the changes in domain structure observed (Fig. 3) can be*
*unequivocally attributed to the role of substrate clamping. In both the clamped and*
*freestanding cases, the SRO layer is the same thickness (30nm), so, while the SRO layer*
*may play a small role in epitaxially constraining the BFO, its effect is present in both the*
*clamped and freestanding cases. Therefore, any differences we observe between these*
*two cases can be attributed to the substrate clamping alone, and not to the SRO layer.”*

2. Although the switching kinetic has been simulated by phase field, the
detailed switching procedure regarding the behaviours of 71/109 domains in
the clamped film and 180 domains in the freestanding films needs to be
discussed. These switching behaviours should also be linked to the observed
change of ferroelectric switching voltage and dynamics.

The referee raises a good point. Our prior work (in *Nature* **516**, 370–373 (2014)),
was carried out in the lab of Professor Huey (U. CONN) using a specialized
setup as described below:

“The switching dynamics of the BiFeO₃ (100 nm)/ SrRuO₃ (8 nm) structure
were measured by time dependent, dual-frequency PFM using an Asylum
Research Cypher in ambient conditions. To activate polarization switching, a
3.515 V direct-current offset was applied to a conductive probe (Nanosensors
NCHR) during scanning along the BiFeO₃ direction. This was superimposed
with a 2 V peak-to-peak amplitude, applied at both the normal cantilever
contact resonance frequency of approximately 1.8 MHz, as well as at a lateral
resonance of approximately 1 MHz. This generates spectrally distinct
piezoresponses along the [001]_p and [100]_p directions that are analysed with a
multi-frequency lock-in amplifier (Zurich Instruments HF2LI). The phase
signals are used to determine the domain orientation along <111>_p. 85
consecutive frames were recorded to resolve polarization-switching dynamics
with a temporal resolution per pixel per image frame of ~40 μs.”

Unfortunately we do not have this capability in our lab; we will be reaching out
to Prof. Huey for an complete experimental study of the switching pathways in
a membrane vs. clamped film. In our time-resolved pulsed switching

measurements, we only observe current from the out-of-plane component of
polarization, so are unable to resolve two-step switching in those experiments.
Finally, we have initiated experiments using time-resolved second harmonic
generation techniques to explore the possibility of capturing the switching
pathways in real time.

We note, however, that the results in the present manuscript are
self-consistent and valid, since they are focused on *what* happens to the
energetics and dynamics of the switching process. Our experimental results
show that there is a reduction in switching time and energy, and the presented
theoretical calculations offer further details and context. Our experimental data
presented in this paper (**Fig. 3, 4**) is agnostic to the switching pathway.

**Reviewer #2** (Remarks to the Author):

This paper is dedicated to a deliberate experiment to understand the role of
oxygen octahedral in switching process of BiFeO₃ ferroelectric domain. In the
experiment the author gains ~60% switching speed by controlling the oxygen
tilting. Moreover, the significance of the octahedral tilting is demonstrated
elegantly by the phase field simulation. It provides us a brand new point of
view to dig into the basic mechanism of the domain switching dynamics. It is
an interesting work and worthy of publication in Nat. Commun, and I
recommend that the paper can be accepted for publication, subject to
answering the following questions.

Q1: When people talk about lattice dynamics, they always mention phonon
behaviors in crystal, which are momentum and frequency dependent. The
author mentioned this as well in the introduction, however they did not discuss
it anymore in the following part of the paper. It would be great if the author
could try to discuss the phonon behavior in the case of “tilting clamping” of
oxygen octahedral. Otherwise, the title of lattice dynamics is somehow
misleading.

We thank the referee for this insightful observation. Indeed, our ultimate
long-term goal is to be able to reveal and understand the interplay between
lattice dynamics and dipolar dynamics. As the referee points out, lattice
dynamics does involve the behavior of phonons, which are frequency and
momentum dependent. The substrate will undoubtedly modify the phonons of
a thin-film, altering their energetics and dispersion characteristics. For example,
a perovskite substrate that does not contain any oxygen-octahedral tilts will
probably soften (and reduce the equilibrium amplitude of) the tilts of a BFO film,
which will in turn impact (probably facilitate) ferroelectric switching. In contrast,
a perovskite substrate presenting rigid oxygen octahedral tilts may harden the
corresponding phonons of the film, and maybe act as a “built-in field” of sorts
for the BFO tilts; this will most likely result in slower switching, potentially
modifying the switching path.

Even if we restrict ourselves to simple considerations as those just mentioned
– ignoring subtler effects related to phonon dynamics and their interplay with
polarization and strains –, it is clear that this is an exceedingly difficult and
subtle problem. Resolving these issues will require systematic theoretical
investigations (via atomistic lattice dynamics) that provide insight into these
complex interactions. It will also require a systematic experimental
investigation of a collection of film+substrate choices addressing the different

scenarios. Admittedly, the present article is a first step in this direction, and
additional work is required to flush out further details.

In the revised manuscript we have inserted an extra paragraph to make this
point and clarify what we mean by “lattice dynamics” in relation to switching.

Please see an excerpt below:

*“It is widely believed that the fundamental limit on ferroelectric switching speed is thus set by*
*the phonon dispersion relation, and specifically the group velocity for acoustic phonons in the*
*system (i.e., speed of sound), which sets a limit on how fast the lattice can respond. For films*
*clamped to a substrate, the substrate will undoubtedly modify the phonons of the thin-film,*
*altering their energetics and dispersion characteristics. For example, a perovskite substrate*
*that does not contain any oxygen-octahedral tilts will probably soften (and reduce the*
*equilibrium amplitude of) the O6 tilts (ferrodistortive order parameter) of a BFO film, which will*
*in turn impact, and likely facilitate, ferroelectric switching. In contrast, a perovskite substrate*
*presenting rigid oxygen octahedral tilts may harden the corresponding phonons of the film, and*
*may act as a “built-in field” of sorts for the BFO tilts; this will most likely result in slower*
*switching, and potentially even modify the switching pathway. Clearly, the substrate, and the*
*mechanical boundary conditions it imposes, plays a critical role in influencing the lattice*
*dynamics of the film.*

*Even if we restrict ourselves to simple considerations as those just mentioned – ignoring subtle*
*effects related to phonon dynamics and their interplay with polarization and strains – it is clear*
*that this is an exceedingly difficult problem. As such, we devise a tractable set of theoretical*
*calculations and experiments that aims to answer a question that addresses how lattice*
*dynamics influence polarization reversal, namely, what is the role of the substrate in*
*influencing ferroelectric switching? We begin by considering the clamping effect ... ”*

Having said this, we are not wedded to the specific title of the manuscript and
we are happy to change it to, for example, “The role of strain and octahedral tilt
clamping in ferroelectric switching” if the referee deems it necessary.

Q2: The details of the “tilting clamping” are not demonstrated in the methods
part. Would the author add these details?

We thank the referee for pointing this out. We provide more information on our
thermodynamic calculations and add the details of tilting clamping in the
Methods section. Please see details of the changes here:

*In Methods Section:*

*“Thermodynamic calculations*

The calculations of free-energy profiles are performed using in-house code for solving the
system of equations $\frac{\partial \varphi_i}{\partial t} = -L_\varphi \frac{\partial F}{\partial \varphi_i}$, [53] where F is defined by Equation 1 (Main text), φ_i is
the order parameter (polarization, rotation or strain tensor components) and L_φ is the kinetic
coefficient describing the rate at which φ approaches its equilibrium value. Since, in these
simulations, we are only interested in the equilibrium values of the order parameters and not in
their time evolution, the values of L_φ do not play role and can be set to 1.

We perform the calculations of the free-energy profiles for the polarization switching path in
which 109° out-of-plane switch (P_y and P_z polarization components are reversed) is followed
by the 71° in-plane switch (P_x component is reversed). This path is chosen based on PFM
experiments on BFO/SRO heterostructures previously reported [12], in which the initial
polarization switching event follows the aforementioned step sequence. In the following we will
only discuss the 109° out-of-plane switch, though the same considerations can be applied to
71° in-plane switch.

For the freestanding BiFeO₃ membrane we assume that during the polarization switching
process all order parameters can freely evolve and adapt to the instantaneous values of the
switching polarization components. For example, for 109° out-of-plane switch, at each value of
$P_y = -P_z$ between -0.7 and 0.7 C/m², we allow P_x as well as of all the components of θ and the
strain tensor, to evolve to their preferred values as dictated by $\frac{\partial \varphi_i}{\partial t} = -L_\varphi \frac{\partial F}{\partial \varphi_i}$. Then, we use the
relaxed values of P , θ and strain to compute the energy of the system, at each step of the
switch. For completeness, the evolution of all order parameters with varied P_y is shown in **Fig.**
**S12** of the Supporting Information (for a phase-field parameter set).

“Strain Clamped” vs. “Strain + Tilt Clamped”

In the simulations of the films clamped by the substrate, we consider three possible clamping
effects. First, we assume that the substrate clamps only ϵ_{11} , ϵ_{22} and ϵ_{12} components of the
strain tensor (“strain clamping” case). To reflect this in our simulations, we fix ϵ_{11} , ϵ_{22} and
ϵ_{12} to their equilibrium values corresponding to the initial direction of the polarization and to
the considered set of model parameters (either DFT or phase-field). Then, similarly to the
case of the freestanding membrane, for 109° out-of-plane switch we vary $P_y = -P_z$ components
of the polarization between -0.7 and 0.7 C/m² and for each P_y value we optimize P_x , θ_x , θ_y , θ_z
as well as the unclamped components of the strain tensor. The evolution of the order
parameters with varying P_y is presented in **Fig. S12** of the Supporting Information together
with that of freestanding case.

Next, we consider the possibility that the presence of the substrate can have an additional
clamping effect on other order parameters, such as FeO₆ octahedral tilts (“strain+tilt
clamping”). In particular, we study two cases, to which we refer in the following as “weak
strain+tilt clamping” and “strong strain+tilt clamping”. In the case of weak strain+tilt clamping,
in addition to ϵ_{11} , ϵ_{22} and ϵ_{12} , we fix the non-switching components of polarization and tilts

(P_x and θ_x for 109° out-of-plane switch) to their initial equilibrium values. Then, we vary
 $P_y=-P_z$ components and for each considered P_y value we optimize θ_y , θ_z as well as the
 remaining components of the strain tensor (ϵ_{33} , ϵ_{13} and ϵ_{23}). The evolution of all order
 parameters with P_y is presented in **Fig. S13** of the Supporting Information. To simulate the
 case of strong strain+tilt clamping, we first identify the equilibrium values of all order
 parameters for both sets of the Landau potential coefficients (DFT and phase-field). Then, we
 vary $P_y=-P_z$ and $\theta_y=-\theta_z$ between their equilibrium values corresponding to up and down
 polarization directions while we keep P_x , θ_x , as well as ϵ_{11} , ϵ_{22} and ϵ_{12} fixed to their initial
 values. For each value of P_y and θ_y we relax the unclamped components of the strain tensor
 (ϵ_{33} , ϵ_{13} and ϵ_{23}). The evolution of the order parameters is shown in **Fig. S14** of the
 Supporting Information.”

Supporting Information Additions

**Figure S12. Extended Strain Clamping.** Evolution of the polarization, octahedral tilts and
 strain tensor components during the 109° polarization switching. The curves denoted as
 “clamped” correspond to the strain clamping case (ϵ_{11} , ϵ_{22} and ϵ_{12} are fixed to their
 equilibrium values corresponding to the initial direction of the polarization while all the other
 order parameters are allowed to relax.)

**Figure S13 Extended Weak Strain + Tilt Clamping.** Evolution of the polarization, octahedral
 tilts and strain tensor components during the 109° polarization switching. The curves denoted
 as “clamped” correspond to the “weak strain+tilt clamping” case (non-switching components of
 polarization and tilts (P_x and θ_x) as well as ϵ_{11} , ϵ_{22} and ϵ_{12} components of the strain tensor
 are fixed to their initial equilibrium values. For each value of P_y only θ_y , θ_z and the unclamped
 components of strain tensor are allowed to relax).

**Figure S14 Extended Strong Strain + Tilt Clamping.** Evolution of the polarization,
 octahedral tilts and strain tensor components during the 109° polarization switching. The
 curves denoted as “clamped” correspond to the “strong strain+tilt clamping” case ($P_y = -P_z$ and
 $\theta_y = -\theta_z$ are varied between their equilibrium values corresponding to up and down polarization
 directions while P_x , θ_x , as well as ϵ_{11} , ϵ_{22} and ϵ_{12} are fixed to their initial values. For each
 value of P_y and θ_y only ϵ_{33} , ϵ_{13} and ϵ_{23} components of the strain tensor are allowed to
 relax).

Q3: The coercive voltage is determined by many effect; it is not convincing to
 identify the experimental clamping effect as it lies somewhere between strain
 clamping and strain+tilt clamping from the energy point of view, although it
 seems straight forward.

We thank the referee for the comment. Indeed, the coercive field can be
 impacted by many effects such as defects, grain boundaries, depolarizing
 fields, etc. Here, by using samples grown in identical conditions and then
 removing the substrate after growth, we control for any such effects and are
 able to focus solely on the role of the substrate clamping in dictating switching.
 On the matter of strain clamping vs. strain + tilt clamping, it is impossible
 to disentangle the two effects in our experiment since we do not have selective
 control of the type of clamping by the substrate (*i.e.*, strain and strain + tilt
 clamping). In simulation, however, we do have selective control and show that
 the case of strain + tilt clamping results in a dramatically higher energy barrier
 for switching to occur. In the updated manuscript, we deepen our analysis,
 including varying degrees of clamping, notably adding so-called “weak strain +
 tilt clamping”, which explores an intermediate level of rigidity. This theoretical
 insight provides context for experiment and allows us to conclude, as the
 referee correctly states, that the experimentally observed changes lie between

the cases of purely strain clamping vs strain + tilt clamping. In effect, our
combined theoretical work and experiments indicate that while clamping does
inhibit the rotation of the oxygen octahedral tilt axis, it does not fully clamp it,
*i.e.*, force oxygen octahedral tilts to remain in their initial orientation before
switching. In fact, in including our new findings on “weak strain+tilt clamping”,
we show agreement between experiment and theory where the true effects of
clamping lie between “strain clamping” and “strong strain + tilt clamping”
(formally referred to as “strain + tilt clamping”, precisely as the referee has
noted. This is a novel and significant conclusion and we have made changes
to the manuscript to highlight this finding throughout. We will list here a few key
changes (albeit not a complete list):

*“We expect the true clamping effects to lie between “strain clamped” and “strong strain + tilt*
*clamped” cases, which serve as limiting cases of clamping effects. To quantify such an*
*intermediary degree of clamping, we consider a third type of clamping termed “weak strain +*
*tilt clamping.” In this case, (Fig. 2b, 2c, 2g, 2h) we fix the non-switching tilt components, but*
*allow switching tilt components to adapt freely to the change in polarization. In this case we*
*see a smaller reduction in the energy barrier upon releasing the film from the substrate, which*
*more accurately describes our experimental data, discussed below.”*

*“The coercive voltage is a measure of the energy required to switch the polarization, and our*
*observed coercive voltage ratio (~40%) between free-standing and clamped films indicate that*
*the clamping effect lies somewhere between the limiting cases of strain clamping and strong*
*strain + tilt clamping, and is most accurately described by weak strain + tilt clamping (Fig. 2).*
*This is an important finding, and deserves special attention. Strain clamping alone, where the*
*mechanical effect of the substrate only inhibits switching via the strain order parameter, is*
*insufficient in explaining the dramatic reduction in energy observed experimentally. The*
*substrate clamping plays an additional role, namely that the mechanical constraints imposed*
*also inhibit variation of oxygen octahedral tilting. Only when both effects are considered, can*
*we explain the significant reduction in switching energy observed. Furthermore, the extreme*
*case of strong strain + tilt clamping predicts a reduction in energy larger than that observed.*
*This indicates that while clamping effects during switching inhibit changes in octahedral tilting,*
*they do not completely prevent these changes from occurring.”*

*Update to Figure 2:*

**Figure 1. Thermodynamic calculation of switching free energy for BFO.** a. (e.) and b. (f.) show 109°,
 out-of-plane, and 71°, in-plane, switching energy landscapes, respectively, calculated using Landau
 coefficients obtained from DFT (used in the phase-field model, Supporting Information Section 1) for the
 strain-clamped and membrane cases. c. (g.) and d. (h.) show 109° and 71° double well potentials,
 respectively, calculated using the Landau potential from DFT (from the phase-field model) for strain + tilt
 clamped and membrane cases. In all panels, the “membrane” curves (blue) correspond to a film free of
 constraints, i.e. all order parameters are free to adapt to the switching polarization. To obtain the
 “clamped” (solid orange curves) results in panels a., b., e., and f., the in-plane strains are held fixed,
 modeling the effect of strain clamping from the substrate. “Clamped (Weak)” (solid orange) curves in c.,
 and d., (g., and h.) represent switching potentials derived from DFT parameters (phase-field parameters),
 but subject to so-called “weak strain + tilt clamping” constraints, where, additionally, all non-switching
 polarization and tilt components are held fixed. “Clamped (Strong)” (dashed orange) curves in c., and d.,
 (g., and h.) show switching potentials derived from DFT parameters (phase-field parameters), but subject
 to so-called “strong strain + tilt clamping” constraints, where, all the non-switching polarization and
 ferrodistortive components are held fixed and the switching components of tilts are linearly interpolated
 between the values corresponding to the minima of the free-energy curves. Percentages listed are
 reductions in maximum energy barrier for membrane vs. clamped films in each scenario. Calculations
 correspond experimentally to the thickest, fully relaxed, films. In the “strain clamping” case, the in-plane
 strains are fixed to their values in the initial state, before the 109° switch occurs. Hence, the fixed strains
 cannot adapt to the new polarization state after the 109° switch and, therefore, the resulting state has
 higher energy than the initial one. This results in the asymmetric shape of the free-energy curves (in the
 case of “strain+tilt clamping”, in addition to in-plane strains, the non-switching polarization and tilt
 components are fixed as well). Note that this effect is more pronounced in the simulations using the
 phase-field model parameter set (panels e-h) compared to the DFT one (a-d). This occurs because the
 relative magnitude between the C_{ijkl} parameters (entering the strain-related terms in Eq. 1) and the other
 model parameters in the phase-field set is higher compared to the DFT set which leads to the stronger
 predicted clamping effect.

Q4: The calculated free energy switching pathway for BFO is neither the same
 as the calculation in Physical Review B 98,085107 (2018) nor Nature 516, 370
 (2015). Would the author explain the difference?

Let us first note the experimental evidence for considering the particular
switching path discussed in this work. As it has been demonstrated in Nature
516, 370 (2014), the polarization switching in BiFeO₃/SrRuO₃ heterostructure
grown on a DyScO₃ substrate occurs in two steps: either 71° in-plane switch is
followed by 109° out-of-plane switch or vice versa (109° out-of-plane switch
followed by 71° in-plane switch). Extended data Figure 2 of Nature 516, 370
(2014) shows that the 71° in-plane switch occurs as the first switching event in
55.4% of the sample under the positive PFM tip bias ($V > 0$) and in 30.6% under
the negative one ($V < 0$). In turn, 109° out-of-plane switch occurs as the first
switching event in 40.7% of the sample under $V > 0$ and 53.3% of the sample
under $V < 0$. After carefully analyzing the results of the time-resolved PFM
switching experiment provided by the authors of Nature 516, 370 (2014), we
noticed that the region of the sample which switches first during the
experiment undergoes the 109° out-of-plane switch followed by 71° in-plane
switch. Therefore, we performed the calculations of the switching-energy
landscape for this switching path.

As for Physical Review B 98,085107 (2018) and the theoretical section of
Nature 516, 370 (2014), those were focused on identifying the lowest energy
path for a full polarization reversal, using density functional theory alone. The
essential conclusion of those works is perfectly consistent with our present
analysis: a multi-step switch where the tilts follow the polarization is the
lowest-energy solution. The details differ, indeed, but this is not surprising,
given that the way in which the lowest-energy path is searched (e.g., by
employing the nudged-elastic band method to find the connection between
given initial and final points) differs from the relatively simpler approach used
here (i.e., interpolation of the switching polarization components and relaxation
of the remaining – not clamped – order parameters).

Additionally, we added an extra note explaining the choice of the switching
path in the Methods section. Please see an excerpt here:

*“We perform the calculations of the free-energy profiles for the polarization switching path in
which 109° out-of-plane switch (P_y and P_z polarization components are reversed) is followed
by the 71° in-plane switch (P_x component is reversed). This path is chosen based on PFM
experiments on BFO/SRO heterostructures previously reported [12], in which the initial
polarization switching event follows the aforementioned step sequence. In the following we will
only discuss the 109° out-of-plane switch, though the same considerations can be applied to
71° in-plane switch.”*

Q5: The significance of octahedral tilting in the ferroelectric switching behavior
of BFO was demonstrated in previous papers, for example L.Bellaiche et. al.

Physical Review Letters 99,227602 (2007), L.Q.Chen, et. al. Physical Review
B 98,085107 (2018) and J. Hlinka, et. al. Physical Review B 96, 174110 (2017).
The timescale of experiment in Fig. 4 is in nano-second, while it is in
pico-seconds in Fig. 5. Although the discrepancy of the timescale is possibly
stems from the limitation of grid size in phase field simulation, it brings forth the
concern that the experiment is quasi-static process comparing to the phase
field simulation. In which case, the contribution of octahedral tilting is well
known from static calculations.

We thank the referee for the comments and have updated the manuscript in
order to more accurately describe the ways which the present work goes
beyond previous findings, especially pertaining the role of substrate clamping.

Physical Review B 98,085107 (2018) gives a detailed look into the energetics
of two switching pathways in BiFeO₃. The difference between pathways in that
work is whether or not the oxygen octahedral rotations (and therefore the
Dzyaloshinskii-Moriya vector and canted magnetic moment) switch. The key
findings regarding energetics of switching are as follows: in the case of what
the authors call “deterministic” switching (*i.e.*, the switching of P always
switches the oxygen octahedral rotations) the authors observe a 25%
reduction in energy barrier vs the so-called “non-deterministic” pathway (*i.e.*,
the switching P does not always result in reversal of oxygen octahedral
rotations). While one can conclude from these findings that the deterministic
switching pathway is preferred, in line with our presented pathways, our
findings differ in that they, for the first time, assign an energetics contribution of
oxygen octahedral rotations with and without mechanical constraints from the
substrate. This is an important distinction, as our work quantitatively
establishes how mechanical clamping from the substrate impacts oxygen
octahedral rotation during ferroelectric switching, and does not address the
question of deterministic vs non-deterministic switching. Furthermore, while we
offer quantitative findings on the energy barrier of switching when including
mechanical clamping of the oxygen octahedra, we go further, performing
time-resolved measurements and phase-field calculations to study how
oxygen octahedra play a dynamic role *during* switching.

In Physical Review B 96, 174110 (2017), the authors explore *ab-initio*
calculations as a means of detailing LGD energy contributions from strain,
polarization, and oxygen octahedra, but do not mention switching, or switching
pathways. They develop a holistic theory, though they do not address how the
various energy costs contribute to the dynamic switching process. Similarly,
Physical Review Letters 99,227602 (2007), is another example of
development of a detailed and highly successful effective Hamiltonian for the
BFO system, including, but not limited to, oxygen octahedral rotations order

parameters. There, the authors do not study switching nor the role of
mechanical constraints' impact.

These papers are now cited, and addressed in the introduction. Please see a
excerpt here:

*“BFO, which follows a two-step polarization switching pathway, consisting of out-of-plane*
*(109°) and in-plane (71°) steps (Fig 1c.), with its ferrodistorive oxygen octahedral tilts following*
*the ferroelectric polarization [12], [27], is therefore an ideal candidate for studying the role of*
*clamping in impacting the switching of several, coupled, primary order parameters. Previous*
*theoretical works have developed highly successful theories for the equilibrium energetics of*
*the BFO system, including effects from oxygen octahedral tilting [28], [29], though have not*
*addressed how substrate clamping influences such energetics (nor dynamics) of the switching*
*process. We theoretically study varying degrees of clamping, and introduce the notion of*
*“strain + tilt clamping” where the substrate influences not only the ferroelectric and strain order*
*parameters, but also, importantly, the oxygen octahedral tilts.”*

Note on timescales of experiment vs calculation:

We thank the referee for this comment, and have added additional discussion
in the manuscript to address this point. The observed experimental switching
time is convoluted with free-charge dynamics in the measurement circuit. It
has been shown previously (Phys. Rev. Lett. 125, 067601 (2020)) that by
modifying the free-charge dynamical timescales via device geometry, one can
control the observed switching time of the ferroelectric. This is because the
RC-time of the measurement circuit imposes limits on how fast one can deliver
charge to facilitate polarization switching. In the presented time-resolved
phase field simulation, no such constraints are imposed. This is the source of
the discrepancy that the referee has noted, and is not because of quasi-static
switching. Further, in the present work, we normalize by RC-time for each
measured device (**Fig 4d.**) to appropriately account for the convolution of
ferroelectric switching dynamics (set by the BFO) with the free-charge
dynamics (set by the measurement circuit). By performing such a
normalization, we are able to make meaningful comparisons across material
systems (e.g. freestanding vs clamped films), and demonstrate agreement
between experiment and theory in terms of changes in switching time upon
release from the substrate. For example, theory predicts that switching times
for freestanding films should constitute a ~75% reduction from clamped films
(**Fig 5**), while we observe a reduction of ~63% (**Fig 4**). Please see an excerpt of
changes to the manuscript here:

*“It is known that the dynamical timescale of free-charge in the measurement circuit, namely*
*RC (resistance×capacitance) time, plays a significant role in the ferroelectric switching times*
*observed in macroscopic device structures at these timescales[20] because the RC-time of the*

*measurement circuit imposes limits on how fast one can deliver charge to facilitate polarization*
*switching. In order to account for any such effects, we normalized all measured switching*
*times to the measured RC-time for each device (Fig. 4d) [20], [25], thereby enabling us to*
*make meaningful comparisons across material systems (e.g. freestanding vs clamped films).*
*Clear decreases in switching time persist even after such normalization, verifying that the*
*observed changes are from the mechanical clamping and not changes to extrinsic-circuit*
*parameters.”*

*“The apparent discrepancy between the absolute values of switching times observed*
*experimentally (~10s ns) and predicted theoretically (~10s ps) is resolved by accounting for*
*the convolution of free-charge dynamics in the experimentally observed value. Corroborating*
*the true dynamic nature (opposed to quasi-static) of the experiment is the remarkable*
*agreement between reductions in switching time predicted (~63% for the 60nm film, and ~75%*
*as predicted by simulation).”*

**Reviewer #3** (Remarks to the Author):

The manuscript under consideration deals with experimental and numerical
investigations of ferroelectric switching in thin films made of bismuth ferrite
(BiFeO₃, BFO). Goal is to reduce the energy barriers required to electrically
switch domain patterns of the films as well as to reduce the associated
switching times. Key idea is to relieve the BFO film from its substrate and
therewith to allow for energetically more favorable switching paths. The
authors analyze the switching of both clamped films (with substrate) and
freestanding films (without substrate) experimentally and numerically. On the
experimental side, piezoresponse force microscopy, X-ray-diffraction and
pulsed ferroelectric measurements are employed to study in detail the
microstructure and ferroelectric response of the films (e.g., lattice parameters,
ferroelectric domain structures, hysteresis). On the numerical side, phase-field
modeling is used to study the evolution of microstructures under applied
electric fields (ab-initio calculations are also presented, but only briefly
addressed). Based on experiments and numerical simulations, the authors
quantify the substrate's impact on coercive fields, saturation polarization and
associated switching times. It is shown that freestanding films could
significantly reduce the necessary energy barriers and the times related to
ferroelectric switching. Furthermore, by numerically comparing classical
clamping (termed "strain clamping" in the manuscript) and additional freezing
of the oxygen octahedral tilts, it is found that tilting has a large impact on the
ferroelectric switching of the films.

The authors document a number of very interesting and novel findings on a
theoretically and practically relevant subject. By combination of detailed
experimental and numerical studies, they have been able to study in-depth
various complex phenomena in BFO films that, according to my knowledge,
haven't been analyzed/demonstrated to that extent before. The methods used
are at the state of the art and mostly well documented (see below for additional
comments/questions in that regard) and the conclusions are sound.

Below, I list a number of remarks and questions of which I hope that they will
prove useful for the authors to further improve the quality of their manuscript.

1) While certainly clear to most of the readers, the authors may want to
introduce BiFeO₃/BFO as bismuth ferrite in the abstract/introduction.

We thank the referee for the comment and have updated the manuscript
accordingly. We now refer to bismuth ferrite (BiFeO₃) in the abstract. Please
see an excerpt here:

*"... combining thermodynamic calculations, experiments, and time-resolved phase-field*
*simulations on both freestanding bismuth ferrite (BiFeO₃) membranes and films clamped..."*

2) In the abstract, the statement "[...] it is found that removing the constraints
imposed by the requirement of macroscopic continuity of deformation (also
known as the von Mises criterion in the deformation of solids [...])" may be
misleading. In fact, "removing the constraints imposed by the requirement of
macroscopic continuity of deformation" could be understood as if the authors
were taking into account macroscopic discontinuities, such as cracks.
However, the two scenarios considered ("freestanding" and "clamped") still
constitute continuous configurations. Also the notion of the "von Mises criterion
in the deformation of solids" could be further explained (or removed) as it
doesn't seem to be a classical criterion employed in solid mechanics for the
present purposes.

We thank the referee for this observation. Our intention was to highlight the
role of macroscopic continuity of deformation from the substrate to the film.
Indeed, macroscopic discontinuities such as cracks can be one way to
alleviate the role of clamping. In the two scenarios considered here, *i.e.*
freestanding and clamped, although the films are continuous, the deformation
of the ferroelectric layer in the clamped case is constrained by its attachment
to the substrate, which is much thicker. We agree with the referee the von
Mises criterion is not needed, and it has been removed from the manuscript.

*"...it is found that by removing the constraints imposed by mechanical clamping from the*
*substrate we can realize a ~40% reduction of the switching voltage and a consequent ~60%*
*improvement in the switching speed..."*

3) A useful and interesting outcome on the side of numerical modeling is
certainly the significance of the oxygen octahedral tilt as an additional order
parameter when modeling BFO. Since the established and widely used
Ginzburg-Landau theories for ferroelectrics (including BFO) often employ as
an order parameter the polarization only, the authors may want to elaborate a
bit more on the ferrodistorive effect; perhaps by adding some more
explanatory information to the supplementary material of the manuscript. In
that context, it would also be appreciated if the authors could motivate the
used model in a more detailed way. From what I understand, the tilts represent

rotations. What kind of reference frame are the angles related to? How do the
authors treat the periodicity of rotations w.r.t. 2π and its impact on gradient
energy and interpolation? How do the authors motivate the corresponding
equation for the ferrodistorive strain? How about the emergence of couple
stresses?

Furthermore, please provide a literature reference related to the model written
down in Equation 1.

We thank the referee for the comment, and provide a detailed discussion for
each of the points below.

(i) **Elaboration on ferrodistorive effect and motivation for our**
**model:**

It is impossible to exaggerate the importance of the octahedral tilts in
BFO, particularly when it comes to discussing ferroelectric switching
and its impact in the magnetic structure of the material. It is well
established (*Phys. Rev. B* **71**, 060401(R) (2005)) that the direction
of the magnetization in BFO is determined by the axis of the
octahedral tilts, and magnetization sign being controlled by the
sense of rotation of the O_6 groups. Thus, deterministic control of the
magnetization requires deterministic control of the tilts, so it is critical
to understand how polarization switching affects the tilting pattern.
Further, as shown in the literature (*Nature* **516**, 370–373 (2014)) and
further evidenced in the present work, the tilts (which constitute a
primary order parameter of BFO, as robust as the polarization itself
and strongly coupled to it) play an active role in determining the
ferroelectric switching path, with in turn determines the evolution of
the magnetization. It is thus evident that any realistic theory of BFO
(in particular, one addressing switching behavior) must include the
tilts explicitly. Our theory does so.

(ii) **Additional information on oxygen octahedral rotations:**

- a. We have added an additional figure to the Supporting
Information, **Figure S15** which highlights the coordinate system
and orientation of rotations. Please see changes here:

We added a **Figure S15** in the Supplementary material illustrating the atomic
displacements corresponding to polar distortion and FeO_6 octahedral rotations
along/around one of the crystallographic directions.

**Figure S15.** Ionic displacement patterns corresponding to (a) polar distortion (along z axis) and (b) FeO_6 octahedral
 rotations (around z axis). Arrows show only the directions of the ionic displacements and do not reflect their relative
 amplitudes. Bi cations are highlighted in blue, Fe cations in green and O anions in red.

b. In our simulations, the antiphase tilts are assumed to have zero amplitude in the high-symmetry cubic structure. Finite values correspond to distortions of the cubic lattice, and our models predict (in agreement with density functional theory and experimental observations) that the tilts reach their most stable (lowest-energy) configurations for rotation angles of about ± 10 degrees. If the angles keep increasing in magnitude, the energy quickly ramps up, making it all but impossible for the simulated system to reach such configurations. Hence, while our polynomic Landau-type potentials do *not* account for the 2π periodicity in the rotation angles, this is not an issue as, in practice, the tilts are confined to the interval between -20 and 20 degrees. Let us stress that our models share this feature with all polynomic models used in solid-state physics, including the usual perturbative treatment of harmonic and anharmonic lattice dynamics.

(iii) **Ferrodistorive strain and coupled stress:**

a. The ferrodistorive eigenstrain comes from the lattice distortion due to the rotation of the oxygen octahedra. In our phase field simulation, the high symmetry phase is chosen to be the high temperature cubic phase, thus due to symmetry, the relationship

between the ferrodistortive eigenstrain and the oxygen
octahedral tilt is quadratic, connected through the rotostrictive
tensor. This is an analogy of the electrostrictive tensor.
b. In our simulation stress is always calculated by multiplying the
stiffness with elastic strain. The elastic strain is solved from the
mechanical equilibrium equation given an eigenstrain distribution.
As far as the mechanical coupling between polarization and
ferrodistortive order is concerned, though there are no cross
terms that directly couple their eigenstrain, such coupling does
occur indirectly, through the mechanical effect. Changing the
value of polarization's or oxygen octahedral tilt's eigenstrain will
cause changes in the solution of the stress and strain distribution,
which will then cause changes in both sets of order parameters.
(iv) **Literature reference for equation 1:**
a. The reference (Phys. Rev. B 90, 220101(R) (2014)) is now
included in the manuscript immediately following equation 1.

We have added additional discussion to the manuscript highlighting this
important finding of how clamping effects the oxygen octahedral tilts, and, in
fact, deepened our analysis to detail how varying degrees of oxygen
octahedral clamping (see "strong strain + tilt clamping" vs "weak strain + tilt
clamping") impact switching. Please see a number of excerpts of changes
here:

*"Here, we present a detailed theoretical and experimental analysis of the role of substrate*
*clamping in influencing ferroelectric switching in the proper ferroelectric, BFO. While all*
*clamped thin-film ferroelectrics are subject to clamping constraints from the substrate,*
*clamping effects can play a larger role in inhibiting ferroelastic switching pathways, which are*
*understood to be lower energy for a variety of systems [10], [11]. BFO, which follows a*
*two-step polarization switching pathway, consisting of out-of-plane (109°) and in-plane (71°)*
*steps (Fig 1c.), with its ferrodistortive oxygen octahedral tilts following the ferroelectric*
*polarization [12], [27], is therefore an ideal candidate for studying the role of clamping in*
*impacting the switching of several, coupled, primary order parameters. Previous theoretical*
*works have developed highly successful theories for the equilibrium energetics of the BFO*
*system, including effects from oxygen octahedral tilting [28], [29], though have not addressed*
*how substrate clamping influences such energetics (nor dynamics) of the switching process.*
*We theoretically study varying degrees of clamping, and introduce the notion of "strain + tilt*
*clamping" where the substrate influences not only the ferroelectric and strain order parameters,*
*but also, importantly, the oxygen octahedral tilts. We show that "strain clamping" alone*
*(ignoring the role of the substrate in clamping the ferrodistortive order) is insufficient in*

*explaining the changes to energetics and dynamics of switching in BFO freestanding vs*
*clamped films, which we observe in experiment..”*

*“There exist other ways in which the substrate can impact switching, namely by clamping*
*additional order parameters (such as octahedral tilts) beyond strain. This can arise, for*
*example, by a mismatch in roto-strictive coefficients between the film, electrode, and substrate.*
*To quantify this effect, we introduce “strain + tilt clamping” (and varying degrees, i.e., weak vs*
*strong, thereof) and compute the free-energy profiles while imposing additional constraints on*
*some order parameters in our simulations. In the limiting case of “strong strain + tilt clamping”*
*(Fig 2g, 2h) we fix all non-switching polarization and tilt components (as well as $\epsilon_{11}, \epsilon_{22}$ and*
*ϵ_{12}) to their equilibrium values corresponding to the initial polarization direction. Moreover,*
*we do not fully relax the switching tilt components, but interpolate them between the values*
*corresponding to the minima of the free energy curves (Methods). Only strains $\epsilon_{33}, \epsilon_{13}$,*
*and ϵ_{23} are allowed to relax following the variation of the other order parameters. One can*
*see that clamping of other degrees of freedom immediately results in a greater difference*
*between the membrane and clamped cases (about 64% for both the 109° and 71° steps). We*
*expect the true clamping effects to lie between “strain clamped” and “strong strain + tilt*
*clamped” cases, which serve as limiting cases of clamping effects. To quantify such an*
*intermediary degree of clamping, we consider a third type of clamping termed “weak strain +*
*tilt clamping.” In this case, (Fig. 2b, 2c, 2g, 2h) we fix the non-switching tilt components, but*
*allow switching tilt components to adapt freely to the change in polarization. In this case we*
*see a smaller reduction in the energy barrier upon releasing the film from the substrate, which*
*more accurately describes our experimental data, discussed below.”*

*“The coercive voltage is a measure of the energy required to switch the polarization, and our*
*observed coercive voltage ratio (~40%) between free-standing and clamped films indicate that*
*the clamping effect lies somewhere between the limiting cases of strain clamping and strong*
*strain + tilt clamping, and is most accurately described by weak strain + tilt clamping (Fig. 2).*
*This is an important finding, and deserves special attention. Strain clamping alone, where the*
*mechanical effect of the substrate only inhibits switching via the strain order parameter, is*
*insufficient in explaining the dramatic reduction in energy observed experimentally. The*
*substrate clamping plays an additional role, namely that the mechanical constraints imposed*
*also inhibit variation of oxygen octahedral tilting. Only when both effects are considered, can*
*we explain the significant reduction in switching energy observed. Furthermore, the extreme*
*case of strong strain + tilt clamping predicts a reduction in energy larger than that observed.*
*This indicates that while clamping effects during switching inhibit changes in octahedral tilting,*
*they do not completely prevent these changes from occurring.”*

4) When describing the characteristic switching process of BFO, the authors
could add more clarity by stating what plane they are referring to (i.e., to what
entity "out-of-plane" and "in-plane" are related to).

We thank the referee for the comment and have made changes accordingly.
 We are referring to the ferroelectric polarization vector, and ‘in-plane’ refers to
 a change in polarization in the plane of the sample surface (the $\langle 001 \rangle_{pc}$ plane),
 while out of plane refers to the direction normal to the plane of the sample. For
 further clarity, the process is shown in Figure 1c, which we have updated to
 highlight the meaning of out-of-plane and in-plane switches. We have included
 an updated discussion as follows:

“BFO typically undergoes 180° switching via a two-step process (Fig. 1c): a 109° switch
 (where the out-of-plane polarization component reverses together with one in-plane
 component) followed by a 71° switch (where the remaining in-plane component reverses), or
 vice versa.”

 **Figure 2. Role of Clamping.** a. Transmission electron microscope (TEM) image of SRO/BFO/SRO heterostructure. b.
 Schematic highlighting significant mechanical constraints imposed by the substrate compared with the freestanding
 film. c. SRO/BFO interface schematic showing ferrodistortive oxygen octahedra rotations and switching pathway (109°
 out-of-plane followed by 71° in-plane) for BFO films.

 5) The sentence "We first calculate the energy associated with
 "strain-clamping" (Fig. 2a-d), i.e. to obtain the curves labeled "clamped", we
 impose that strains epsilon_11, epsilon_22 and epsilon_12 be fixed at the
 values corresponding to the initial state" should be reformulated. Same applies
 to "We obtain similar switching paths the membrane case, [...]". The latter
 statements shall also conclude with " 71° step". Later on the same page: "With
 regards the change".

We thank the referee for this suggestion and have reformulated the
corresponding paragraphs:

*“Next, we consider the case of so-called “strain clamping”. In order to separate clamping*
*effects from effects of epitaxial misfit strain, we assume the film is thick enough so that it is fully*
*relaxed in its rhombohedral ground state. The fully relaxed nature of the film does not, however,*
*mean that it is free to deform; on the contrary, it is still clamped and, as dictated by the*
*substrate, energetically favors maintaining its original state. To obtain the free energy profiles*
*labeled “clamped” in Fig. 2a(b), 2e(f) we vary P_y and P_z for 109° switch (P_x component for*
*71° switch) while keeping the strains $\epsilon_{11}, \epsilon_{22}$ and ϵ_{12} fixed to their equilibrium values*
*corresponding to the initial polarization direction (before the initial 109° out-of-plane switch). All*
*the other order parameters are allowed to relax following the polarization switching process*
*(Methods). One can see that strain clamping leads to slightly increased energy barriers*
*compared to the freestanding case, about 6% for the 109° step and 20% for the 71° step.*
*Notably, the results obtained for the first-principles Landau potential (Fig. 2a and 2b) and the*
*phenomenological model (Fig. 2e and 2f) are essentially equivalent with regards to the change*
*in activation energy barriers, highlighting the consistency of both methods. Additional*
*free-energy calculations for the prototypical ferroelectrics PTO and BTO are presented for*
*clamped and membrane cases (Fig S1). These calculations, consistent with previous*
*experimental work [25], show a similar reduction in switching energy for freestanding*
*membranes compared to clamped films suggesting a broad applicability of the role of strain*
*clamping effects in ferroelectric switching.*

*There exist other ways in which the substrate can impact switching, namely by clamping*
*additional order parameters (such as octahedral tilts) beyond strain. This can arise, for*
*example, by a mismatch in roto-strictive coefficients between the film, electrode, and substrate.*
*To quantify this effect, we introduce “strain + tilt clamping” (and varying degrees, i.e., weak vs*
*strong, thereof) and compute the free-energy profiles while imposing additional constraints on*
*some order parameters in our simulations. In the limiting case of “strong strain + tilt clamping”*
*(Fig 2g, 2h) we fix all non-switching polarization and tilt components (as well as $\epsilon_{11}, \epsilon_{22}$ and*
*ϵ_{12}) to their equilibrium values corresponding to the initial polarization direction. Moreover,*
*we do not fully relax the switching tilt components, but interpolate them between the values*
*corresponding to the minima of the free energy curves (Methods). Only strains $\epsilon_{33}, \epsilon_{13}$,*
*and ϵ_{23} are allowed to relax following the variation of the other order parameters. One can*
*see that clamping of other degrees of freedom immediately results in a greater difference*
*between the membrane and clamped cases (about 64% for both the 109° and 71° steps for the*
*phase-field parameter set). We expect the true clamping effects to lie between “strain*
*clamped” and “strong strain + tilt clamped” cases, which serve as limiting cases of clamping*
*effects. To quantify such an intermediary degree of clamping, we consider a third type of*
*clamping termed “weak strain + tilt clamping.” In this case, (Fig. 2b, 2c, 2g, 2h) we fix the*
*non-switching tilt components, but allow switching tilt components to adapt freely to the*

*change in polarization. In this case we see a smaller reduction in the energy barrier upon*
 *releasing the film from the substrate, which more accurately describes our experimental data,*
 *discussed below.”*

6) Figure 1 (in particular 1c): As the authors later refer to the Cartesian
 coordinates of the polarization vector, a coordinate system shall be added
 already here.

We thank the referee for the feedback, and have included a coordinate system
 in an updated version of the figure.

*Figure 3. Role of Clamping. a. Transmission electron microscope (TEM) image of SRO/BFO/SRO heterostructure. b.*
 *Schematic highlighting significant mechanical constraints imposed by the substrate compared with the freestanding*
 *film. c. SRO/BFO interface schematic showing ferrodistorptive oxygen octahedra rotations and switching pathway (109°*
 *out-of-plane followed by 71° in-plane) for BFO films.*

7) Figure 2: (i) A comment on why we observe unsymmetric landscapes for
 most of the clamped cases would be appreciated. (ii) In e and f, we don't seem
 to have reached an energy minimum yet. (iii) Caption: "This results in higher
 energy barriers, observed".

We thank the referee for pointing this out. In the “strain clamping” case, the
 in-plane strains are fixed to their values in the initial state, before the 109°
 switch occurs. Hence, the fixed strains cannot adapt to the new polarization
 state resulting from the 109° switch and, therefore, the resulting state has

higher energy than the initial one. This results in the asymmetric shape of the
free-energy curves shown in Fig. 2. (In the case of “strain+tilt clamping”, in
addition to in-plane strains, the non-switching polarization and tilt components
are fixed as well.) Note that this effect is more pronounced in the simulations
using the phase-field model parameter set (2b,c,e,f) compared to the DFT one
(a,d). This occurs because the relative magnitude between the C_{ijkl} parameters
(the strain-related terms in Eq. 1) and the other model parameters in the
phase-field set is higher compared to the DFT set which leads to the stronger
predicted clamping effect.

We added an explanation of the asymmetric free-energy curves in the caption
of Fig. 2. Please see updates here:

**Figure 4. Thermodynamic calculation of switching free energy for BFO.** a. (e.) and b. (f.) show 109°,
out-of-plane, and 71°, in-plane, switching energy landscapes, respectively, calculated using Landau
coefficients obtained from DFT (used in the phase-field model, Supporting Information Section 1) for the
strain-clamped and membrane cases. c. (g.) and d. (h.) show 109° and 71° double well potentials,
respectively, calculated using the Landau potential from DFT (from the phase-field model) for strain + tilt
clamped and membrane cases. In all panels, the “membrane” curves (blue) correspond to a film free of
constraints, i.e. all order parameters are free to adapt to the switching polarization. To obtain the
“clamped” (solid orange curves) results in panels a., b., e., and f., the in-plane strains are held fixed,
modeling the effect of strain clamping from the substrate. “Clamped (Weak)” (solid orange) curves in c.,
and d., (g., and h.) represent switching potentials derived from DFT parameters (phase-field parameters),
but subject to so-called “weak strain + tilt clamping” constraints, where, additionally, all non-switching
polarization and tilt components are held fixed. “Clamped (Strong)” (dashed orange) curves in c., and d.,
(g., and h.) show switching potentials derived from DFT parameters (phase-field parameters), but subject
to so-called “strong strain + tilt clamping” constraints, where, all the non-switching polarization and
ferrodistorive components are held fixed and the switching components of tilts are linearly interpolated
between the values corresponding to the minima of the free-energy curves. Percentages listed are
reductions in maximum energy barrier for membrane vs. clamped films in each scenario. Calculations
correspond experimentally to the thickest, fully relaxed, films. In the “strain clamping” case, the in-plane
strains are fixed to their values in the initial state, before the 109° switch occurs. Hence, the fixed strains
cannot adapt to the new polarization state after the 109° switch and, therefore, the resulting state has
higher energy than the initial one. This results in the asymmetric shape of the free-energy curves (in the
case of “strain+tilt clamping”, in addition to in-plane strains, the non-switching polarization and tilt
components are fixed as well). Note that this effect is more pronounced in the simulations using the
phase-field model parameter set (panels e-h) compared to the DFT one (a-d). This occurs because the
relative magnitude between the C_{ijkl} parameters (entering the strain-related terms in Eq. 1) and the other
model parameters in the phase-field set is higher compared to the DFT set which leads to the stronger
predicted clamping effect.

8) The authors state that they are "using now a complete Ginzburg-Landau
potential resolved via the Landau-Khalatnikov dynamical equation [46]". What
precisely does that mean? First, the cited paper proposes a multiferroic model,
including magnetic and magnetoelectric effects. Second, why is the thus far
used model not sufficient anymore?

We thank the referee for the comments. We have corrected our citation to
Physical Review B 90, 220101(R) (2014). The referee is correct in stating that
the previous reference discusses a multiferroic model, and we have corrected
our error in the updated reference, aforementioned. By use of the word
"complete" we mean that the model used in the phase-field simulation includes
gradient terms, which were not included in the single domain calculation of
switching energy barriers.

To avoid confusion, we have removed the reference to the Landau-Khalatnikov
equation in favor of a more clear "time dependent Ginzburg-Landau equation"
and moved detailed discussion to Methods. We have also updated the text to
make it more clear.

Please see updates as follows:

"... domain structure predicted by phase-field simulations, calculated using the same
Ginzburg-Landau potential [33] as that used to calculate free-energy switching landscapes
(Fig. 1) and including gradient terms to account for domain configuration evolution (Methods)."

In Methods:

"**Phase-field simulation.** The current phase-field model for ferroelectric free standing film is
an extension to our previous model for bulk and epitaxial thin film simulations[32], [33], [51],
[54], [55] in which we use the spontaneous polarization $\mathbf{p}=(p_1, p_2, p_3)$ and oxygen octahedral
tilt $\boldsymbol{\theta} = (\theta_1, \theta_2, \theta_3)$ as the order parameters to represent the ferroelectric domain structure
evolution that is coupled to the elastic and electric state of the whole system. A temporal
evolution of the order parameter's spatial distribution can be obtained by solving the
time-dependent Ginzburg-Landau equation[56], [57]:

$$\frac{\partial P_i}{\partial t} = -L_P \frac{\delta F_{tot}}{\delta P_i}, \quad (i=1, 2, 3) \quad (2)$$

$$\frac{\partial \theta_i}{\partial t} = -L_\theta \frac{\delta F_{tot}}{\delta \theta_i}, \quad (i=1, 2, 3) \quad (3)$$

in which L_p and L_θ are the kinetic coefficients, t is time, uppercase P and θ represent the
 spatial distribution of the order parameter p and θ at each simulation grid point, F_{tot} is the
 total free energy of BFO membrane, given as,

$F_{tot} = \iiint_{V_{BFO}} (f_{land} + f_{grad} + f_{elec} + f_{elast}) dV$, and $f_{land} = \alpha_{ij} p_i p_j + \alpha_{ijkl} p_i p_j p_k p_l + \beta_{ij} \theta_i \theta_j +$

$\beta_{ijkl} \theta_i \theta_j \theta_k \theta_l + t_{ijkl} p_i p_j \theta_k \theta_l$ represent the local free energy densities of Landau energy, which
 includes the landau energy for polarization, oxygen octahedral tilt and the coupling terms,

$f_{grad} = \frac{1}{2} g_{ijkl} p_{i,j} p_{k,l} + \frac{1}{2} \kappa_{ijkl} \theta_{i,j} \theta_{k,l}$ represents the gradient energy, which includes the gradient

energy for both polarization and oxygen octahedral tilt, $f_{elec} = -\frac{1}{2} E_i (\epsilon_0 \epsilon_{ij} E_j + p_i)$ represents

the electrostatic energy, and $f_{elast} = \frac{1}{2} C_{ijkl} (\epsilon_{ij} - \epsilon_{ij}^0) (\epsilon_{kl} - \epsilon_{kl}^0)$ represents the elastic energy,

in which the eigenstrain is determined by the order parameters $\epsilon_{ij}^0 = \lambda_{ijkl} \theta_k \theta_l + Q_{ijkl} p_k p_l$.

There are two differences between our free-standing film and our epitaxial thin film model[33],

both of which pertain to solving for the mechanical equilibrium state. First, the clamped thin

film has a displacement free bottom and traction-free top surface, while for the free-standing

film, both the top and bottom surface are set as traction-free boundaries, as shown in equation

(4) owing to the nature of the freestanding membrane,

$$\begin{cases} \sigma_{i3}|_{z=0} = 0 \\ \sigma_{i3}|_{z=h_f} = 0 \end{cases}, \quad (i=1,2,3) \quad (4)$$

where h_f means the membrane thickness. Second, the in-plane macroscopic strain in

clamped thin film is controlled by the misfit of the substrate, while for the free-standing

film, the in-plane macroscopic strain is set to be the average eigenstrain calculated from

the current order parameter distribution, since overall the free-standing film is in a

stress-free state,.

We start our simulation from random noise for both the polarization and oxygen

octahedral tilt in a 128 nm * 128 nm * 30 nm thick clamped thin film with 0.4%

compressive biaxial misfit and 0 applied electric field. The system is relaxed to an

equilibrium state as shown in Figure 3(d). Then the same domain structure is used as an

input to the free-standing film simulation without applied electric field, from which we get

the domain structure of Figure 3(e). Next, starting from these equilibrium state, we first

*apply an instantaneous negative electric potential, -10V, on both films' top surface*
*(bottom grounded) to pole the system fully upwards, then remove the potential to relax*
*the system back to equilibrium. The domain state at this stage is 71 stripe domains in the*
*clamped film and single domain in the free-standing film. Finally, an instantaneous 13V*
*electric potential is applied on both films' top surface (bottom still grounded) to switch the*
*domains downwards, during which process we keep track of the energy evolution as*
*shown in Figure 5.*

*Our simulations are periodic in the x and y directions. In the clamped film case, there are 6*
*grids of air layer above and 10 grids of substrate layer below the film along the z direction.*
*While in the free-standing film case, there are 6 grids of air layer above and 0 grids of*
*substrate layer below the film."*

9) Figure 3: (i) Could the authors provide a color scale for the shown plots? (ii)
What are the spatial dimensions of the numerically simulated regions? (iii)
What is the thickness of the film considered in the numerical simulations? (iv)
The authors have employed a periodic representation the film by using
periodic boundary conditions. This shall be stated explicitly in the manuscript.
Furthermore, did the authors perform a study w.r.t. the size of the periodic
region? (v) The orientations of the stripe domains is different in (b) and (d):
Could the authors explain? (vi) Caption: "(a)" and "(b)" shall be replaced with
"(b)" and "(c)".

*We the thank the referee for the comments and provide a detailed list of*
*answers below:*

*(i) We have updated the figure to include a color scale. (see updated figure*
*below)*

*(ii) The simulated region is 128nm x 128nm. We have added a scale bar for*
*clarity. (see updated figure below)*

*(iii) The film/membrane thickness is 30nm in phase field simulation.*

*(iv) The boundary conditions is stated more clearly in the Methods and*
*Supplementary Information. We are using a fourier spectral method to solve*
*the mechanical equilibrium equation. We state this explicitly in the SI. Details*
*of how we enforce the thin film boundary conditions onto the system are*

outlined in Y. L. Li *et. al.* Acta Materialia, **50** 2 (2002). We did perform studies
varying the size of the simulation region. For smaller size simulations, there
will be fewer domain strips and it will be much harder to get the stripe in two
directions, like those shown in Figure 3d. We make our simulation size as large
as possible, subject to the constraint that larger simulation sizes result in very
long wait times for the calculation to complete. The current simulation size of
128 nm * 128 nm * 30 nm is at the upper limit of feasibility.

Please see changes to the manuscript here:

In Methods:

*“... Our simulations are periodic in the x and y directions. In the clamped film case, there
are 6 grids of air layer above and 10 grids of substrate layer below the film along the z
direction. While in the free-standing film case, there are 6 grids of air layer above and 0
grids of substrate layer below the film. ...”*

In Supporting Information:

**“Section 3: Numerical Details**

*We are using a semi-implicit Fourier spectral method [5] to evolve the time dependent
Ginzburg-Landau equation. The whole system is in a periodic condition with air/substrate layer
above and below the ferroelectric material in the Z direction. For a film simulation, since the
grid points outside the film are either air or substrate, the order parameters are initialized with
non-zero values within the film and fixed at 0 outside the film.*

*The mechanical equilibrium, $\sigma_{ij,j} = 0$, is solved using a Fourier spectral solver, which will give
1018 us the stress and strain distributions that are necessary to calculate the elastic driving force for
order parameters. The mechanical system is periodic in the X and Y directions. To implement
the boundary conditions for clamped and free-standing film along Z direction, we used a
superposition method that is explained in details in our previous publication[6].*

*The electrostatic equation, $\epsilon_0 \epsilon_{ii} \phi_{,ii} = P_{i,i}$, is also solved using a Fourier spectral solver, which
will give us the electric field distribution that is needed for the electric driving force to the
polarizations. The electrostatic system is periodic in the X and Y directions. For both the
clamped and free-standing film cases, we are using a short-circuit boundary conditions for the
film top and bottom surface/interface along Z direction, which means the electric potential is*

fixed on the top and bottom. The bottom surface/interface is always grounded (i.e., electric
potential equals 0). We change the electric potential on the top surface between negative, zero,
and positive, depending on whether we are poling, relaxing or switching the system.”

(v) The orientation of the sample is different in b. and d. We have made this
clear by including domain sample crystallographic axes labels in the figure.
(see updated figure below)

(vi) We thank the referee, the wrong labels in the caption are fixed.

10) The statement "The saturation polarization of the freestanding membrane
is lower than that of the clamped film, consistent with our observed reduction in
c/a ratio (Fig. 3) and quasi-static hysteresis loops (Fig 4a,b)." is more or less a
repetition of what has just been noted a few lines earlier.

We thank the referee for the suggestion and have removed the sentence.

11) The abbreviation "RC" may be introduced.

We have included an updated discussion of RC times. We have included a
snippet of the changes here:

“It is known that the dynamical timescale of free-charge in the measurement circuit, namely
RC (resistance×capacitance) time, plays a significant role in the ferroelectric switching times
observed in macroscopic device structures at these timescales[20] because the RC-time of the
measurement circuit imposes limits on how fast one can deliver charge to facilitate polarization
switching.”

12) "In order to account for any such effects, we normalized all switching times
to the measured RC-time constant for each device (Fig. 3d)": The number of
the figure shall be replaced with "4".

We thank the referee for catching the typo and have updated the text
accordingly.

Updated text:

"... we normalized all measured switching times to the measured RC-time for each device (Fig.
4d)"

13) Figure 4 (caption): "free-standing films 25nm samples".

We have updated the caption to make it more clear. Updated caption:

"*Switching dynamics as a function of applied voltage for the clamped and free-standing 25nm*
*films*"

14) When introducing the model used for the numerical simulations, the
authors refer to "(Methods, [49])". The cited models seem to be different,
however. What kind of model has been used? Wouldn't it be more convenient
so select only one model and to deduct any simulations based on that (see
also remark 8).

We have fixed both the references mentioned in referee comments 8 and 14.
We do use the same phase-field model in this paper, and are not mixing
several models together. We apologize for the confusion.

15) The way in which parts of the numerical simulations were performed could
be described in a bit more detailed way. For example, when saying "All
simulations start from an equilibrium domain state with polarization pointing
downwards, a positive voltage is then applied on top to switch the polarization
upwards": What is the potential at the bottom? How was that potential at the
top applied (ramped, instantaneous)? What is the influence of the chosen
boundary conditions on the dynamics of the different domain structures? What
are the initial conditions for the tilts?

We thank the referee for the comments. We have added more details of the
simulation procedure in Methods and included details of numerical calculations
in the Supporting Information.

The bottom surface/interface is always grounded (*i.e.*, electric potential equals
0 V). The potential applied on the top surface is instantaneously applied and is

maintained constant throughout the simulation. Oxygen octahedral tilts are
initialized to a random state at each simulation grid point. We then let the
system evolve to an equilibrium state, which we refer to as a “relaxation”
process. We then begin the poling. The relaxation and poling process are
described in detail in Methods. The tilts have a similar pattern as the
polarization (*i.e.*, 71° stripe domain for clamped film and single domain for
free-standing film) before we apply any poling voltage.

The question of how boundary conditions influence the dynamics of varied
domain structures is an interesting one. A proper study would entail varying
electric and elastic boundary conditions for many different domain types (71°,
109°, 180°) and then performing dynamic evolution under an applied voltage.
This is a significant undertaking, and beyond the scope of the present work.
The aim of the current study is to establish changes in dynamics between
clamped and unclamped films. We do not attempt to produce an exhaustive
study of the dynamics different domain structures subject to different boundary
conditions, though future work may address this directly.

Please see changes to the manuscript here:

Updates to Methods:

*“We start our simulation from random noise for both the polarization and oxygen octahedral tilt*
*in a 128 nm * 128 nm * 30 nm thick clamped thin film with 0.4% compressive biaxial misfit and*
*0 applied electric field. The system is relaxed to an equilibrium state as shown in Figure 3(d).*
*Then the same domain structure is used as an input to the free-standing film simulation*
*without applied electric field, from which we get the domain structure of Figure 3(e). Next,*
*starting from these equilibrium state, we first apply an instantaneous negative electric potential,*
*-10V, on both films’ top surface (bottom grounded) to pole the system fully upwards, then*
*remove the potential to relax the system back to equilibrium. The domain state at this stage is*
*71 stripe domains in the clamped film and single domain in the free-standing film. Finally, an*
*instantaneous 13V electric potential is applied on both films’ top surface (bottom still grounded)*
*to switch the domains downwards, during which process we keep track of the energy evolution*
*as shown in Figure 5.”*

Updates to SI:

**“Section 3: Numerical Details**

*We are using a semi-implicit Fourier spectral method [5] to evolve the time dependent*

*Ginzburg-Landau equation. The whole system is in a periodic condition with air/substrate layer*
*above and below the ferroelectric material in the Z direction. For a film simulation, since the*
*grid points outside the film are either air or substrate, the order parameters are initialized with*
*non-zero values within the film and fixed at 0 outside the film.*

*The mechanical equilibrium, $\sigma_{ij,j} = 0$, is solved using a Fourier spectral solver, which will give*
*us the stress and strain distributions that are necessary to calculate the elastic driving force for*
*order parameters. The mechanical system is periodic in the X and Y directions. To implement*
*the boundary conditions for clamped and free-standing film along Z direction, we used a*
*superposition method that is explained in details in our previous publication[6].*

*The electrostatic equation, $\epsilon_0 \epsilon_{ii} \phi_{,ii} = P_{i,i}$, is also solved using a Fourier spectral solver, which*
*will give us the electric field distribution that is needed for the electric driving force to the*
*polarizations. The electrostatic system is periodic in the X and Y directions. For both the*
*clamped and free-standing film cases, we are using a short-circuit boundary conditions for the*
*film top and bottom surface/interface along Z direction, which means the electric potential is*
*fixed on the top and bottom. The bottom surface/interface is always grounded (i.e., electric*
*potential equals 0). We change the electric potential on the top surface between negative, zero,*
*and positive, depending on whether we are poling, relaxing or switching the system.”*

16) The style of Fig. 11 could be adjusted to the style of Fig. 5 as both show
similar results.

We have updated Supplementary Figure S11 to match the style of Figure 5.

17) In the conclusion: "all coupled order parameters much be understood".

*The manuscript has been updated to correct this typo. Now: "all coupled order*
 *parameters must be understood"*

18) In the methods section, under "phase-field simulation": (i) The " θ "
 indicating the tilt is a vector and shall thus be printed in boldface. (ii) "that is
 couple to the elastic" (iii) "of the order parameter spatial distribution". (iv) In the
 first two equations: In the partial derivative of the polarization w.r.t. time, the "t"
 should be on the same level as the partial sign. (v) In the first two equations:
 The Ginzburg-Landau theory is local in the sense that "f" should appear on the
 right-hand side rather than "F_tot". (vi) In the first two equations: The authors
 are using the same mobility parameter "L" in both equations. Is that true? How
 are these parameters adjusted, in particular for the ferrodistorive part? (vii) "L
 is the kinetic coefficient related to the domain wall mobility": What domain wall
 mobility? (viii) "F_tot" does not contain any dielectric energy. Why? I would
 expect that it is of major relevance here. (ix) "landan". (x) "f_grad" misses "1/2".
 (xi)

"f_elect" should read "f_elec" (as before). (xii) "f_elastic" should read "f_elast"
 (as before). (xiii) The sentence "There are two differences between the current
 free-standing film and our previous epitaxial thin film model [32], all of which
 are in the elastic part" could be misleading as the "elastic part" has just been

associated with the energy. (xiv) The numbering of Equation "2" may be
adjusted. (xv) In Equation 2: "z=0" instead of "z=1"?

We thank the referee for the detailed comments. We have outlined our
responses and changes below.

We have made changes to the manuscript to reflect fixes to points (i), (ii), (iii)
and (iv).

(v) The P and θ in this equation mean the spatial distribution of the
polarization and the tilt rotation rather than the value at a specific location. The
r and t in the parenthesis which may have caused confusion are now removed.
We now use upper case P and Θ to label the spatial distribution of polarization
and oxygen octahedral tilt, and lower case p and θ to represent the
polarization and oxygen octahedral tilt at a specific location.

(vi) In our program, the L for polarization and oxygen octahedral tilt are
separate parameters. This is made more clear in the updated Methods section.

When interested in the final state, both kinetic coefficients can be set to 1
because the equilibrium state is independent of the evolution to said state.

(vii) Domain wall mobility is the equivalent concept of the mobility in a classical
Allen-Cahn equation, for ferroelectrics, it means how fast the domain wall
migrates under electric field. To avoid confusion, we remove this term since it
does not provide any additional insight in the context of our manuscript.

(viii) The coupling between electric field and polarization is considered in the
f_{elec} term $f_{elec} = -(1/2E_i)(\epsilon_0\epsilon_{ij}E_j + p_i)$ (Appl. Phys. Lett. 81, 427 (2002)).

To avoid confusion, we have updated the equation to include the dielectric
energy. We note, however, that when solving for evolution of the polarization
order parameter, we will take a partial derivative with respect to p to get the
driving force for p . In this case, the dielectric component will drop out.

Points ix, x, xi, xii, have been fixed in the manuscript.

(xiii) We have rewritten this sentence to make it more clear.

We have updated the manuscript to include fixes to points (xiv) and (xv).

Please see the changes here:

***Phase-field simulation.*** *The current phase-field model for ferroelectric free standing film is*
*an extension to our previous model for bulk and epitaxial thin film simulations[32], [33], [51],*
*[54], [55] in which we use the spontaneous polarization $\mathbf{p}=(p_1, p_2, p_3)$ and oxygen octahedral*

tilt $\theta = (\theta_1, \theta_2, \theta_3)$ as the order parameters to represent the ferroelectric domain structure
 evolution that is coupled to the elastic and electric state of the whole system. A temporal
 evolution of the order parameter's spatial distribution can be obtained by solving the
 time-dependent Ginzburg-Landau equation[56], [57]:

$$1204 \quad \frac{\partial P_i}{\partial t} = -L_p \frac{\delta F_{tot}}{\delta P_i}, \quad (i=1, 2, 3) \quad (2)$$

$$1205 \quad \frac{\partial \theta_i}{\partial t} = -L_\theta \frac{\delta F_{tot}}{\delta \theta_i}, \quad (i=1, 2, 3) \quad (3)$$

in which L_p and L_θ are the kinetic coefficients, t is time, uppercase P and θ represent the
 spatial distribution of the order parameter p and θ at each simulation grid point, F_{tot} is the
 total free energy of BFO membrane, given as,

$$1209 \quad F_{tot} = \iiint_{V_{BFO}} (f_{land} + f_{grad} + f_{elec} + f_{elast}) dV, \quad \text{and} \quad f_{land} = \alpha_{ij} p_i p_j + \alpha_{ijkl} p_i p_j p_k p_l + \beta_{ij} \theta_i \theta_j +$$

$\beta_{ijkl} \theta_i \theta_j \theta_k \theta_l + t_{ijkl} p_i p_j \theta_k \theta_l$ represent the local free energy densities of Landau energy, which
 includes the landau energy for polarization, oxygen octahedral tilt and the coupling terms,

$$1212 \quad f_{grad} = \frac{1}{2} g_{ijkl} p_{i,j} p_{k,l} + \frac{1}{2} \kappa_{ijkl} \theta_{i,j} \theta_{k,l}$$
 represents the gradient energy, which includes the gradient

energy for both polarization and oxygen octahedral tilt, $f_{elec} = -\frac{1}{2} E_i (\epsilon_0 \epsilon_{ij} E_j + p_i)$ represents

the electrostatic energy, and $f_{elast} = \frac{1}{2} C_{ijkl} (\epsilon_{ij} - \epsilon_{ij}^0) (\epsilon_{kl} - \epsilon_{kl}^0)$ represents the elastic energy,

in which the eigenstrain is determined by the order parameters $\epsilon_{ij}^0 = \lambda_{ijkl} \theta_k \theta_l + Q_{ijkl} p_k p_l$.

There are two differences between our free-standing film and our epitaxial thin film model[33],

both of which pertain to solving for the mechanical equilibrium state. First, the clamped thin

film has a displacement free bottom and traction-free top surface, while for the free-standing

film, both the top and bottom surface are set as traction-free boundaries, as shown in equation

(4) owing to the nature of the freestanding membrane,

$$1221 \quad \begin{cases} \sigma_{i3}|_{z=0} = 0 \\ \sigma_{i3}|_{z=h_f} = 0 \end{cases} \quad (i=1,2,3) \quad (4)$$

where h_f means the membrane thickness. Second, the in-plane macroscopic strain in

clamped thin film is controlled by the misfit of the substrate, while for the free-standing

film, the in-plane macroscopic strain is set to be the average eigenstrain calculated from

the current order parameter distribution, since overall the free-standing film is in a

*stress-free state,.*

*We start our simulation from random noise for both the polarization and oxygen*
*octahedral tilt in a 128 nm * 128 nm * 30 nm thick clamped thin film with 0.4%*
*compressive biaxial misfit and 0 applied electric field. The system is relaxed to an*
*equilibrium state as shown in Figure 3(d). Then the same domain structure is used as an*
*input to the free-standing film simulation without applied electric field, from which we get*
*the domain structure of Figure 3(e). Next, starting from these equilibrium state, we first*
*apply an instantaneous negative electric potential, -10V, on both films' top surface*
*(bottom grounded) to pole the system fully upwards, then remove the potential to relax*
*the system back to equilibrium. The domain state at this stage is 71 stripe domains in the*
*clamped film and single domain in the free-standing film. Finally, an instantaneous 13V*
*electric potential is applied on both films' top surface (bottom still grounded) to switch the*
*domains downwards, during which process we keep track of the energy evolution as*
*shown in Figure 5."*

19) Supporting Information, Section 1: While formulas are stated in full index
notation, the parameters listed in the table are given in Voigt notation. Please
provide a means to relate them. Furthermore, please provide associated
references.

*We thank the referee for the comment. The conversion between Voigt and*
*tensor notation and references are provided in SI. See excerpt here:*

***"Relationship between Voigt notation and tensor notation:***

$\alpha_1 = \alpha_{11}, \alpha_{11} = \alpha_{1111}, \alpha_{12} = 2\alpha_{1122}$

$\beta_1 = \beta_{11}, \beta_{11} = \beta_{1111}, \beta_{12} = 2\beta_{1122}$

$\lambda_{11} = \lambda_{1111}, \lambda_{12} = \lambda_{1122}, \lambda_{44} = 4\lambda_{1212}$

$t_{11} = t_{1111}, t_{12} = t_{1122}, t_{44} = 2t_{1212}$

$C_{11} = C_{1111}, C_{12} = C_{1122}, C_{44} = C_{1212}$ "

20) Supporting Information, Section 2: It is not clear to me how the authors are
treating the mechanical boundary conditions. Could they describe that in a
more detailed way? In particular, I wonder how they realized the in-plane

stress-free conditions related to the freestanding films during the whole
process of the simulations.

We have included more details about simulation in the SI (we have added an
additional section, Section 3, to this point). Since we are using Khachaturyan's
microelasticity theory to solve for the mechanical equilibrium with boundary
conditions (more details Y. L. Li *et. al.* Acta Materialia, **50** 2 (2002)) to achieve
the in-plane stress-free condition, we update the in-plane component of
homogeneous strain using the average in-plane eigenstrain for every
simulation timestep. The treatment of the mechanical boundary conditions is
laid out in a very detailed fashion in the reference (Y. L. Li *et. al.* Acta Materialia,
**50** 2 (2002)). We have highlighted the key ideas in the SI, though, to avoid
repeating our previous work, we refer the reader to that reference for the full
explanation. Please see an excerpt of the added section (Section 3) here:

*"The mechanical equilibrium, $\sigma_{(ij,j)}=0$, is solved using a Fourier spectral solver, which will*
*give us the stress and strain distributions that are necessary to calculate the elastic driving*
*force for order parameters. The mechanical system is periodic in the X and Y directions. To*
*implement the boundary conditions for clamped and free-standing film along Z direction, we*
*used a superposition method that is explained in details in our previous publication[6].*

*The electrostatic equation, $\epsilon_0 \epsilon_{ij} \phi_{(ii)}=P_{(i,i)}$, is also solved using a Fourier spectral solver,*
*which will give us the electric field distribution that is needed for the electric driving force to the*
*polarizations. The electrostatic system is periodic in the X and Y directions. For both the*
*clamped and free-standing film cases, we are using a short-circuit boundary conditions for the*
*film top and bottom surface/interface along Z direction, which means the electric potential is*
*fixed on the top and bottom. The bottom surface/interface is always grounded (i.e., electric*
*potential equals 0). We change the electric potential on the top surface between negative, zero,*
*and positive, depending on whether we are poling, relaxing or switching the system."*

21) Follow-up comment: The results reported in the paper heavily rely on
numerical simulations. To allow for better reproducibility, I would like to ask the
authors to provide more background information on the used numerical
schemes (theory, algorithms, discretizations, boundary conditions, ...) in the
supplementary material of the manuscript.

We thank the referee for the suggestion and have included additional
discussion in the Supplementary Material for the phase field simulations and in
the Methods section for the thermodynamic calculations. We also described
how the DFT parameter set have been calculated, please see an excerpt from
Section 1 of SI:

*“In order to provide fully ab initio description of the switching energy landscape in freestanding*
*and clamped films of BiFeO₃, we employ the model described by Eq. 1 and the parameter set*
*extracted using DFT calculations. To obtain this parameter set we apply the following*
*procedure: (i) we construct several structural BiFeO₃ polymorphs, optimize their crystal*
*structures using DFT and extract their equilibrium properties (polarization, octahedral tilts,*
*strains and energies); (ii) we derive the analytical expressions for all the model parameters*
*using the conditions $\frac{\partial F}{\partial \varphi_i} = 0$ (where F is defined by Eq. 1 and φ_i is the order parameter) as*
*well as the expressions for the energies of the considered polymorphs; (iii) we use these*
*analytical expressions and the DFT values of polarization, octahedral tilts, strains and energies*
*to compute the values of the model parameters.*

*All DFT calculations are performed using the Vienna Ab initio Simulation Package (VASP) [2].*
*We employ the generalized gradient approximation (GGA+U) for the exchange-correlation*
*potential in the form of Perdew-Burke-Ernzerhof [3] revised for solids (PBEsol) [4]. For more*
*accurate treatment of Fe 3d electrons of we apply Hubbard-U correction U=4 eV. The*
*calculations are performed using the 40-atom supercell of BiFeO₃ (the perovskite unit cell is*
*doubled along x, y and z directions) and 3x3x3 Monkhorst-Pack k-point grid was used for*
*integration over the Brillouin zone corresponding to this supercell. The energy cutoff for the*
*plane wave basis is set to 500 eV. In all simulations we impose the G-AFM order of Fe*
*magnetic moments.”*

We hope these explanations could be appreciated by you.

Thanks again for your comments and suggestions.

Best,

Qiwu Shi

REVIEWER COMMENTS

Reviewer #1 (Remarks to the Author):

The reviewer is convinced by the additional explanation of the manuscript novelty. The quality of the manuscript has been significantly improved. I can now recommend its publication in Nature Communications.

Reviewer #3 (Remarks to the Author):

I would like to thank the authors for their efforts to improve the quality of their paper and for taking into account my remarks. Below, I comment on individual answers of the authors only where necessary. The numbering refers to the numbering of the first review.

8) Thank you very much for the detailed response. (i) There seem to be several typos in the electrostatic energy that shall be corrected. (ii) How can you guarantee quasi-static conditions when you apply the electric voltage in an instantaneous manner? That also doesn't seem to fit to the experimental conditions (Fig. 4). (iii) What do you mean with "n grids of air layer above" and "m grids of substrate layer"?

9-iv) The electrostatic equation given by the authors deserves more explanation. From what I understand, the authors are solving Gauss's law, $\text{div}(\mathbf{D}) = 0$. This is not stated here, however. Why is the trace of the electric permittivity in place?

15) The authors' answer "The question of how boundary conditions influence the dynamics of varied domain structures is an interesting one. [...] We do not attempt to produce an exhaustive study of the dynamics different domain structures subject to different boundary conditions, though future work may address this directly" is a bit disappointing. As the authors state, they are interested in the dynamics of domain evolution, but they have obviously not studied the influence of boundary conditions. I understand that it is beyond the scope of the paper to study many different domain structures, but at least the influence of electric loading times (instantaneous, ramped) should be investigated as the application of boundary conditions may play a major role. I would therefore like to encourage the authors to either perform such studies or at least explain in detail why their specific choice of boundary conditions does produce reliable/physical results that could be compared with the given experiments.

18-v) This is confusing. Why is it necessary to distinguish between \mathbf{P} and \mathbf{p} ? It should be possible to evaluate the polarization field \mathbf{P} at a point and therewith to obtain \mathbf{p} as per its current definition. Then, one could easily write down the evolution law locally without the need for integrals. Currently, it looks as if on the left-hand side you evaluate the local evolution of the polarization field and on the right-hand side you evaluate an integral expression (F_{tot} is the entire energy). Such a statement reminds of nonlocal theories. Is this what you have in mind here?

18-vi) The authors write "When interested in the final state, both kinetic coefficients can be set to 1 because the equilibrium state is independent of the evolution to said state". This statement is not correct in general since the coupled model used by the authors is non-convex and therefore lacks a unique solution.

18-viii) I am irritated by the expression for the electrostatic energy. Shouldn't its negative partial derivative w.r.t. the electric field give the electric displacement? I have taken a look at the cited paper (Appl. Phys. Lett. 81, 427 (2002)) and indeed could find a similar

expression, which however didn't contribute to clarification. How could the authors derive D from the energy and why doesn't it read " $D = \epsilon_0 E + P$ "? Could the authors explain? I am also confused by authors' comment "We note, however, that when solving for evolution of the polarization order parameter, we will take a partial derivative with respect to p to get the driving force for p . In this case, the dielectric component will drop out". The mentioned term may be irrelevant for the evolution equation of the polarization, but it is relevant for Gauss's law that must be solved alongside mechanical equilibrium and the evolution equations. Could you comment?

Response to Referees Letter for the manuscript "The role of lattice dynamics in ferroelectric switching"

Dear Editor and Referees,

Thank you very much for your recent comments concerning our manuscript (NCOMMS-21-31371A) entitled "The role of lattice dynamics in ferroelectric switching". Based on the comments and suggestions received, we have made careful modifications on the original manuscript, point by point. All the amendments in the manuscript have been highlighted in Yellow. The responses to the reviewers' comments/questions are presented as follows:

Summary of changes

We appreciate the reviewer's detailed comments. To accurately address all of the reviewer's comments, we have significantly updated the Phase-field Simulation part of the Methods section in the revised manuscript. In addition, we outline below detailed responses to each of the reviewer's comments.

Reviewer #3 (Remarks to the Author):

I would like to thank the authors for their efforts to improve the quality of their paper and for taking into account my remarks. Below, I comment on individual answers of the authors only where necessary. The numbering refers to the numbering of the first review.

8) Thank you very much for the detailed response.

(i) There seem to be several typos in the electrostatic energy that shall be corrected.

Response: We thank the referee for this comment. We have corrected the typos in electrostatic energy expression. It now reads:

" $f_{elec} = -\frac{\epsilon_0 \epsilon_{ij}^b}{2} E_i E_j - E_i p_i$ is the electrostatic energy, where ϵ_{ij}^b is the background dielectric constant, E_i is the electric field obtained by solving the electrostatic equilibrium equation $\epsilon_0 (\epsilon_{11}^b \frac{\partial^2 \Phi}{\partial x^2} + \epsilon_{22}^b \frac{\partial^2 \Phi}{\partial y^2} + \epsilon_{33}^b \frac{\partial^2 \Phi}{\partial z^2}) = \frac{\partial p_1}{\partial x} + \frac{\partial p_2}{\partial y} + \frac{\partial p_3}{\partial z}$, Φ is the electrical potential"

(ii) How can you guarantee quasi-static conditions when you apply the electric voltage in an instantaneous manner? That also doesn't seem to fit to the experimental conditions (Fig. 4).

Response: We thank the referee for raising this concern. Before we explain our thinking in greater detail, we make the important note that the experimental protocol differs between **Fig 4a,b** and **Fig 4c,d**. Throughout the manuscript the usage of "quasi-static" is limited only to the experimental ferroelectric hysteresis loop data presented in **Fig 4a,b**, which corresponds to a measurement of the energetics of the switching process. In that context, our usage of "quasi-static" refers the application of a 10kHz ramped electric field (triangular wave) which induces switching and maps out the hysteresis loop. This can be considered "quasi-static" because the stimulus is slowly varying in time. In the dynamic measurements (**Fig 4c,d**), a step voltage is applied to the material, and the ferroelectric polarization evolves dynamically (the onset of the stimulus occurs much faster than the switching). In our phase-field calculations (**Fig 5**), we probe the dynamics by using an instantaneously applied electric field, to most closely mimic the experimental conditions and the dynamics observed in **Fig. 4c,d**. To improve the clarity of our phase-field simulations, we provide the referee with additional details. The mechanical stresses and electric fields are obtained by solving two equilibrium equations knowing the order parameters at any given time. As such, one can think of the mechanical stresses and electrostatic fields as responding instantaneously to changes in the order parameters (polarization and octahedral tilts) during the switching process. By using an instantaneously applied electric field in the simulations, we are able to most accurately match the conditions of our experiment of interest.

(iii) What do you mean with "n grids of air layer above" and "m grids of substrate layer"?

Response: We thank the referee for the comment. For the clamped film condition, we have a total of 46 simulation grid points in the Z direction (each grid represents 1nm), among which the bottom 10 layers represent the substrate, the middle 30 layers represent the BFO film, and the top 6 layers

represent air. We have updated the last two paragraphs of the Phase-field simulations part of the methods section accordingly:

*“... a 128 nm * 128 nm * 30 nm (each simulation grid $\Delta x = 1$ nm) thick clamped thin film ...”*

“... The film thickness is 30 nm, or 30 grids, for both the clamped and free-standing film. In the clamped film case, there are 6 grids of air layer above and 10 grids of substrate layer below the film along the z direction. While in the free-standing film case, there are 6 grids of air layer above and 0 grids of substrate layer below the film. ...”

9-iv) The electrostatic equation given by the authors deserves more explanation. From what I understand, the authors are solving Gauss's law, $\text{div}(\mathbf{D}) = 0$. This is not stated here, however. Why is the trace of the electric permittivity in place?

Response: We thank the referee for the comment. We use $D_i = \varepsilon_0 \varepsilon_{ij}^b E_j + p_i$, where we have separated the electric displacement into a spontaneous polarization component, p , and a background component that changes linearly with electric field by excluding the contribution from the ferroelectric soft-mode. Therefore, at electrostatic equilibrium,

$$\begin{aligned} \nabla \cdot \mathbf{D} = 0 &\rightarrow \frac{\partial D_i}{\partial x_i} = \frac{\partial}{\partial x_i} [\varepsilon_0 \varepsilon_{ij}^b E_j + p_i] = 0 \\ &\rightarrow \varepsilon_0 (\varepsilon_{11}^b \frac{\partial^2 \phi}{\partial x^2} + \varepsilon_{22}^b \frac{\partial^2 \phi}{\partial y^2} + \varepsilon_{33}^b \frac{\partial^2 \phi}{\partial z^2}) = \frac{\partial p_1}{\partial x} + \frac{\partial p_2}{\partial y} + \frac{\partial p_3}{\partial z} \end{aligned}$$

This derivation also explains why the electric permittivity appears in the electric energy, since p is spontaneous polarization not total polarization. The permittivity is the background permittivity (A. K. Tagantsev, *Ferroelectrics*, 69:1, 321-323 (1986) ; Alexander K. Tagantsev, *Ferroelectrics*, 375:1, 19-27 (2008) ; Woo, C., *Appl. Phys. A* **91**, 59–63 (2008) ; A. P. Levanyuk, *Ferroelectrics*, 503:1, 94-103 (2016)) which now denoted as ε_{ij}^b to avoid confusion.

15) The authors' answer "The question of how boundary conditions influence the dynamics of varied domain structures is an interesting one. [...] We do not attempt to produce an exhaustive study of the dynamics different domain structures subject to different boundary conditions, though future work may

address this directly" is a bit disappointing. As the authors state, they are interested in the dynamics of domain evolution, but they have obviously not studied the influence of boundary conditions. I understand that it is beyond the scope of the paper to study many different domain structures, but at least the influence of electric loading times (instantaneous, ramped) should be investigated as the application of boundary conditions may play a major role. I would therefore like to encourage the authors to either perform such studies or at least explain in detail why their specific choice of boundary conditions does produce reliable/physical results that could be compared with the given experiments.

Response: We thank the referee for the thoughtful comment. While we present experimental and theoretical evidence for changes in domain structure before and after liftoff from the substrate, this is a change in the ground state of the system, and does not reflect, nor aim to address a different question, namely, the influence of domain-structure-imposed boundary conditions *during* switching. In our manuscript, we had stated:

"... in which we use the spontaneous polarization $\mathbf{p}=(p_1, p_2, p_3)$ and oxygen octahedral tilt $\theta = (\theta_1, \theta_2, \theta_3)$ as the order parameters to represent the ferroelectric domain structure evolution that is coupled to the elastic and electric state of the whole system. A temporal evolution of the order parameter's spatial distribution can be obtained by solving the time-dependent Ginzburg-Landau equation ..."

The use of "domain structure evolution" is misleading because that's not what we're intending to convey in this manuscript. We do recognize that the issue of domain structure evolution and the associated dynamics is an interesting question, though those calculations are ongoing and will take time to complete properly. Therefore, we have updated this section as shown here:

"... in which we use the spontaneous polarization $\mathbf{p}=(p_1, p_2, p_3)$ and oxygen octahedral tilt $\theta = (\theta_1, \theta_2, \theta_3)$ as the order parameters . A temporal evolution of the order parameters can be obtained by solving the time-dependent Ginzburg-Landau equation ..."

The referee's question of electric loading times deserves additional discussion. A ramping of the field will influence the observed timescales of switching, though the effect requires nuanced treatment. As a first step, we performed a new set of simulations, by adding a time-period of linearly increasing electric field, as shown here:

Figure R1. Dynamic elastic, electric, Landau, and total energy with a ramped electrical loading. Orange (blue) curves show energies for clamped (freestanding) films as a function of time. In all panels, black (dashed) curve corresponds to the right axis (applied voltage in Volts) vs. time. The inset shows the difference between the applied voltage in Figure 5 (red) and that applied here (black, dashed).

To the referee's point, the results are indeed interesting. One can readily observe that the timescales of switching did change (slow down) for both the freestanding and clamped films. For example, a comparison with **Figure 5** shows that the switching time for the membrane slowed from ~ 20 ps to ~ 100 ps after slowing the electrical loading of the system. Since ferroelectric switching is an activated process, the ratio of applied electric field to activation field plays a key role in setting the switching speed. By ramping the applied field, this ratio is not constant. As such, the ramp rate (and not only the maximum applied electric field) is a key parameter in setting the switching speed. For example, with a linearly increasing field like that shown in **Fig R1**, the ramped field will more significantly impact the membrane switching process earlier, simply because the activation field of the membrane is lower and is therefore reached

sooner. This can be observed most clearly in the electric energy component of the energy landscape. Therefore, by changing the ramp rate, we can tune the switching speed. It is important to note, however, that in the present manuscript we are less interested in the role of external stimuli, and instead focused on how internal mechanical constraints dictate switching dynamics. By slowing the electrical loading, we are introducing an external tuning parameter which convolutes the desired intrinsic dynamics. For these reasons, we believe it is most fitting to study the case of instantaneously applied fields.

Finally, we refer the referee to our response to question 8 (ii), in which we discuss the validity of our phase-field simulation in mimicking the experimental conditions. We performed phase-field simulations to provide additional, corroborating information to the experimental observations of **Fig. 4c**, in which an instantaneous voltage is applied. Our phase-field simulations give us the ability to “see” an “energy breakdown” during switching of the polarization. This is key information, providing insight into the main aim of our paper, how the *mechanical* boundary conditions and clamping influence the switching speed.

18-v) This is confusing. Why is it necessary to distinguish between P and p ? It should be possible to evaluate the polarization field P at a point and therewith to obtain p as per its current definition. Then, one could easily write down the evolution law locally without the need for integrals. Currently, it looks as if on the left-hand side you evaluate the local evolution of the polarization field and on the right-hand side you evaluate an integral expression (F_{tot} is the entire energy). Such a statement reminds of nonlocal theories. Is this what you have in mind here?

Response: We apologize, we misunderstood the reviewer’s original question (in round 1 of review). In our original version of the manuscript, we did not distinguish between P and p , and have returned to that notation (staying consistent with our usage throughout the manuscript, as p). For clarity, we offer a detailed explanation of the reviewer’s original question, with additional notes to address questions raised in the second review.

Original question: “In the first two equations: The Ginzburg-Landau theory is local in the sense that “ f ” should appear on the right-hand side rather than “ F_{tot} ”.”

We copy the equations in question here:

$$\frac{\partial p_i}{\partial t} = -L_P \frac{\delta F_{\text{tot}}}{\delta p_i}, \quad (i=1, 2, 3) \quad (2)$$

$$\frac{\partial \theta_i}{\partial t} = -L_\theta \frac{\delta F_{\text{tot}}}{\delta \theta_i}, \quad (i=1, 2, 3) \quad (3)$$

The right-hand side is the variational derivative of the total free energy of a spatially inhomogeneous system with respect to the spatial distribution function of the order parameter which describes the inhomogeneous system. In TDGL theory, we are finding the spatial profile of the order parameter which minimizes the total free energy of the complete system. At equilibrium, the total free energy of the system is minimized, and the corresponding order parameter profile is the equilibrium profile. As such, we should use F_{tot} , *i.e.*, the total energy, and not the local value, f . The referee is correct that this is a non-local theory as we are minimizing the energy of the complete system.

18-vi) The authors write "When interested in the final state, both kinetic coefficients can be set to 1 because the equilibrium state is independent of the evolution to said state". This statement is not correct in general since the coupled model used by the authors is non-convex and therefore lacks a unique solution.

Response: We thank the referee for the comment. In the thermodynamic calculation, we are only interested in the equilibrium values of the order parameters and not the trajectory by which it is reached, so the values of L_ϕ are set to 1 in the reduced units (we checked that the choice of L_ϕ does not affect the resulting equilibrium state of the system). We have made changes to the manuscript to reflect this note:

"The calculations of free-energy profiles are performed using in-house code for solving the system of equations $(\partial \phi_i)/\partial t = -L_\phi \partial f / (\partial \phi_i)$, [53] where f is defined by Equation 1 (Main text), ϕ_i is the order parameter and L_ϕ is the kinetic coefficient describing the rate at which ϕ approaches its equilibrium value. Since, in these simulations, we are only interested in the equilibrium values of the order parameters and not the trajectory by which it is reached, the values of L_ϕ are set to 1 in the reduced units (we checked that the choice of L_ϕ does not affect the resulting equilibrium state of the system)."

The primary reason we choose 1 for both kinetic coefficients in the phase-field section is due to the lack of experimental data for us to fit such parameters for BFO. To address this, in the manuscript, we have added the following to the methods section to clarify this issue:

"Owing to the lack of experimental data to which to fit L_p and L_θ , we set both L_p and L_θ to be 1 after normalizing all coefficients to unitless values."

Finally, we have updated Figure 5 and S11 in favor of arbitrary time units to reflect this choice of kinetic coefficient.

18-viii) I am irritated by the expression for the electrostatic energy. Shouldn't its negative partial derivative w.r.t. the electric field give the electric displacement? I have taken a look at the cited paper (Appl. Phys. Lett. 81, 427 (2002)) and indeed could find a similar expression, which however didn't contribute to clarification. How could the authors derive D from the energy and why doesn't it read "D = epsilon_0 E + P"? Could the authors explain? I am also confused by authors' comment "We note, however, that when solving for evolution of the polarization order parameter, we will take a partial derivative with respect to p to get the driving force for p. In this case, the dielectric component will drop out". The mentioned term may be irrelevant for the evolution equation of the polarization, but it is relevant for Gauss's law that must be solved alongside mechanical equilibrium and the evolution equations. Could you comment?

Response: We thank the referee for the comments. We have corrected the typos and now the electrostatic energy is $f_{elec} = -\frac{\epsilon_0 \epsilon_{ij}^b}{2} E_i E_j - E_i p_i$, which should be thermodynamically consistent. One can do the partial derivative with respect to E to get the negative of electric displacement $D_i = \epsilon_0 \epsilon_{ij}^b E_j + p_i^{spontaneous}$, and one can also do the partial derivative with respect to the order parameter p to get the electrical driving force for polarization domain structure evolution. In the reviewer's equation "D = epsilon_0 E + P", "P" is the total polarization, containing both spontaneous polarization, p, as well as a, electric-field-dependent dielectric component. In our phase-field model, p is the order parameter, *i.e.*, spontaneous polarization only, and the linear dielectric component $\epsilon_0 \epsilon_{ij}^b E_j$ does not include the contribution from the ferroelectric soft-mode.

REVIEWER COMMENTS

Reviewer #3 (Remarks to the Author):

I would again like to thank the authors for their time to improve the quality of their paper. Below, I comment on individual answers of the authors only where necessary. The numbering refers to the numbering of the first and second review.

8i) It is a bit frustrating that the details of the electrostatic energy are revealed only step by step. Now, after the second revision, we get to know that this energy contains a background dielectric constant. Could the authors justify the magnitude/selection of that "material parameter" in the present context? How was it determined? Note that --- surprisingly --- the magnitude of the background dielectric constant is not even given in the paper.

8iii) Strictly speaking, you have only one grid that is composed of several (layers of) grid points.

9-iv) Most of the phase field models for ferroelectrics use the total polarization as an order parameter and couple the deformation to that field. Now that the authors select the spontaneous polarization, which typically occurs in piezoelectric materials, their model should come along with piezoelectric coupling terms in the energy. Why are they omitted here? Furthermore, why can the electrostriction due to "background polarization" be neglected?

18-v) According to the authors' answer, the solution of a global problem is automatically the solution of a nonlocal problem. This is not true. The authors are motivated to reconsider their statement and their equations once more. As a starting point, they may want to question why they have an integral on the right hand side but not on the left hand side of their Ginzburg-Landau equation. Isn't the polarization solved on the whole domain, too?

18-vi) This is a very confusing answer. First of all, the use of reduced units is mentioned for the first time, which is again extremely frustrating. Second, in reduced units one is of course free to reduce any of the values to one (as that doesn't change the physics). But this has nothing to do with the remaining argumentation of the authors: In physical units, the mobility constants have certain magnitudes and these of course matter as they have an influence on the dynamics of the whole system. Therefore, the repeated statement of the authors that the final configuration of the system doesn't depend on the choice of these constants is irritating (in particular in the present case, where we have two order parameters in place). I would therefore like to ask the authors to explain in more detail and to list the selected physical values of mobility parameters alongside the remaining parameters in the table of the supplementary material.

Response to Referees Letter for the manuscript "The role of lattice dynamics in ferroelectric switching"

Summary of changes

We appreciate the reviewer's questions and comments. To address all the reviewer's questions, we have updated the Supplementary Information and made the required changes to the Phase-field part of the Methods section. Below are our detailed responses to each of the reviewer's comments.

Reviewer #3 (Remarks to the Author):

I would again like to thank the authors for their time to improve the quality of their paper. Below, I comment on individual answers of the authors only where necessary. The numbering refers to the numbering of the first and second review.

8i) It is a bit frustrating that the details of the electrostatic energy are revealed only step by step. Now, after the second revision, we get to know that this energy contains a background dielectric constant. Could the authors justify the magnitude/selection of that "material parameter" in the present context? How was it determined? Note that --- surprisingly --- the magnitude of the background dielectric constant is not even given in the paper.

Response: It is a fair criticism that we should have provided more details about our solutions to the electrostatic problem and the value for the background dielectric constant in the original manuscript. The background dielectric constant which we consider is the total dielectric constant minus that arising from the ferroelectric spontaneous polarization contribution [1,2,3,4], which is already included in the thermodynamic description of the ferroelectric phase transition. In order to determine the magnitude of the background dielectric constant, we must avoid convolution from the ferroelectric spontaneous polarization. As such, to get an approximate experimental measurement of the background dielectric constant, we fit the "high-field" portion of the polarization vs. voltage plot (**Fig. R1**), at which point the ferroelectric spontaneous polarization is completely saturated. The background dielectric constant is approximately equal to ~50, which is the value used in simulation.

Figure R1. Linear fit showing background dielectric constant (~ 50) of 100nm (a.) and 35nm (b.) freestanding films.

We added the following paragraph to the Supplementary Information:

“Phase-field simulations are performed at room temperature, $T=298\text{K}$. The relative background dielectric constant $\epsilon_{11}^b, \epsilon_{22}^b, \epsilon_{33}^b$ is 50.”

8iii) Strictly speaking, you have only one grid that is composed of several (layers of) grid points.

Response: We have updated the main text as the reviewer suggested, using “grid point” instead of grid to avoid confusion.

“The film thickness is 30 nm, or 30 layers of grid points, for both the clamped and free-standing film. In the clamped film case, there are 6 layers of grid points of air layer above and 10 layers of grid points of substrate layer below the film along the z direction. While in the free-standing film case, there are 6 layers of grid points of air layer above and 0 layer of grid point of substrate layer below the film.”

9-iv) Most of the phase field models for ferroelectrics use the total polarization as an order parameter and couple the deformation to that field. Now that the authors select the spontaneous polarization, which typically occurs in piezoelectric materials, their model should come along with piezoelectric coupling terms in the energy. Why are they omitted here? Furthermore, why can the electrostriction due to “background polarization” be neglected?

Response: We break our responses into sections, answering each of the reviewer’s comments:

“Most of the phase field models for ferroelectrics use the total polarization as an order parameter and couple the deformation to that field.”

It is not true that most of the phase-field models for ferroelectrics use the total polarization as an order parameter. There are two types of phase-field models used for ferroelectrics. The first is the so-called “phase-separation approach” (often used in the mechanics community) [6], while the second formulation applies Landau-Devonshire theory (often used in the materials community) [7-12]. We use the latter in this work. Among those employing the Landau-Devonshire theory, further sub-classification is required. Research groups such as that of Chad M Landis [7] and Kaushik Bhattacharya [8] use *total* polarization as the order parameter, though others, such as the groups of Long-Qing Chen [9], Yue Zheng [10], Anna N. Morozovska [11], and Jiri Hlinka [12], use the spontaneous polarization as the order parameter.

Further, different communities, e.g., the mechanics and materials communities, have somewhat different definitions of “spontaneous polarization.” We feel this is not the right place for us to comment on the preciseness as well as the advantages and disadvantages of different definitions. Instead, based on our understanding, we will just point out the main differences between the phase-field models of ferroelectrics in the mechanics community and those in the materials community. The mechanics community [6] seems to define only the polarization at zero electric field and zero stress as the “spontaneous polarization” and uses the experimentally measured dielectric constant, which includes all contributions, in the electrostatic equilibrium equation. The phase-field models in the materials community define the ferroelectric (spontaneous polarization) polarization at all electric fields and all stresses as the “spontaneous polarization”. Here, all the non-ferroelectric dielectric contributions (*i.e.*, the total dielectric contributions minus the ferroelectric spontaneous polarization contribution) to the dielectric constant are taken into account through the background dielectric constant in the electrostatic equilibrium equation.

“Now that the authors select the spontaneous polarization, which typically occurs in piezoelectric materials, their model should come along with piezoelectric coupling terms in the energy. Why are they omitted here?”

We are not sure whether we fully understand the comment, because all ferroelectrics are piezoelectrics. We believe the confusion could be from the different definitions of the spontaneous polarization in the two types of phase-field models of ferroelectrics discussed in the last paragraph. While piezoelectric coupling terms (odd in the spontaneous polarization) do not explicitly appear in the energy, it is not true that they are not included. The energy must be invariant upon reversal of $+p \rightarrow -p$, and hence any odd terms of spontaneous polarization, should not exist in the energy. The piezoelectric response of BFO comes from the electrostriction due to the development of both spontaneous polarization and strain, and it is included in our elastic strain energy contributions through the relation between spontaneous strain (eigenstrain) and spontaneous polarization. The electrostrictive coefficient Q_{ijkl} and the piezoelectric

coefficient d_{mij} are directly related through $d_{mij} = Q_{ijkl} p_k \frac{\partial p_l}{\partial E_m}$ [13,14], p is the spontaneous polarization, E is the electric field. As such, the piezoelectric effect in BFO is fully counted for in our phase-field model through the electrostrictive effect in our energy function.

“Furthermore, why can the electrostriction due to “background polarization” be neglected?”

To illustrate the contributions from the background dielectric response vs spontaneous polarization to the overall charge: 1.) The induced (background) polarization at a typical applied field of 500kV/cm (**Fig. R1**) is $\approx \epsilon_0 \times 50 \times 500\text{kV/cm} = 2.2\mu\text{C/cm}^2$, where ϵ_0 is the vacuum permittivity. This is $\sim 2\%$ of the spontaneous polarization. Further, we note that the main aim of this work is to address the switching dynamics, during which, a dynamic stress is induced by the switching of the ferroelectric (spontaneous) polarization, in the thin film that is clamped to the substrate. As can be observed (**Fig. R2**), the slope during switching $\sim 16000\epsilon_0$, indicating that the background polarization has a negligible effect on the main conclusions (role of the lattice during switching) results of this paper.

Figure R2. Negligible effect of electrostriction from background polarization during switching. (Switching data for 35nm sample)

18-v) According to the authors' answer, the solution of a global problem is automatically the solution of a nonlocal problem. This is not true. The authors are motivated to

reconsider their statement and their equations once more. As a starting point, they may want to question why they have an integral on the right hand side but not on the left hand side of their Ginzburg-Landau equation. Isn't the polarization solved on the whole domain, too?

Response: We think the reviewer misunderstood the variational derivative on the right-hand side of the equation. It is a local equation, and we solve the same equation at every grid point within the computational domain. It is true that the total free energy of the whole domain is an integral over the whole spatial domain, but the variational derivative given by the Euler-Lagrange equation is local, and hence there is no inconsistency between the left and right-hand side of the equation; both the left and right-hand sides are local. For example,

$$F(\rho) = \int [f(r, \rho(r), \nabla\rho(r))]d^3r$$

which is an integral over all space while the variational derivative

$$\frac{\delta F}{\delta \rho} = \frac{\partial f}{\partial \rho} - \nabla \cdot \frac{\partial f}{\partial \nabla \rho} \quad (18.1)$$

is local, defined at any given point. We employ the variational derivative (Equation (18.1)) explicitly in our calculations: (we show the derivation here for the p order parameter, meaning for all of the symbols are the same as in our manuscript)

$$\begin{aligned} \frac{\partial p_i}{\partial t} &= -L_p \frac{\delta F_{\text{tot}}}{\delta p_i} \\ &= -L_p \left(\frac{\partial (f_{\text{land}} + f_{\text{grad}} + f_{\text{elec}} + f_{\text{elast}})}{\partial p_i} - \nabla \cdot \frac{\partial (f_{\text{land}} + f_{\text{grad}} + f_{\text{elec}} + f_{\text{elast}})}{\partial \nabla p_i} \right) \\ \frac{\partial p_i}{\partial t} &= -L_p \left(\frac{\partial (f_{\text{land}} + f_{\text{elec}} + f_{\text{elast}})}{\partial p_i} - \nabla \cdot \frac{\partial f_{\text{grad}}}{\partial \nabla p_i} \right) \end{aligned}$$

$$\frac{\partial p_i}{\partial t} = -L_p (2\alpha_{ij}p_j + 4\alpha_{ijkl}p_j p_k p_l + 2t_{ijkl}p_j \theta_k \theta_l - E - 2C_{ijkl}(\varepsilon_{ij} - \varepsilon_{ij}^0)Q_{ijkl}p_j - g_{ijkl}p_{k,jl})$$

18-vi) This is a very confusing answer. First of all, the use of reduced units is mentioned for the first time, which is again extremely frustrating. Second, in reduced units one is of course free to reduce any of the values to one (as that doesn't change the physics). But this has nothing to do with the remaining argumentation of the authors: In physical units, the mobility constants have certain magnitudes and these of course matter as they have an influence on the dynamics of the whole system. Therefore, the repeated statement of the authors that the final configuration of the system doesn't depend on the choice of

these constants is irritating (in particular in the present case, where we have two order parameters in place). I would therefore like to ask the authors to explain in more detail and to list the selected physical values of mobility parameters alongside the remaining parameters in the table of the supplementary material.

Response: The reviewer has a valid complaint here since we did not fully explain what we meant by “reduced”. To explain what we meant by reduced unit, let’s rewrite our polarization evolution equation by moving the kinetic coefficient L from the right-hand side of the equation to the left-hand side (we show the case of p here, though identical analysis can be performed other order parameters),

$$\frac{\partial p_i}{\partial(tL_p)} = -\frac{\delta F_{\text{tot}}}{\delta p_i}$$

Therefore, we can absorb the kinetic coefficient into time, by redefining, $t \rightarrow t*L$ as the new time parameter. It is in this sense, that we claim “reduced units” for L since we absorbed the unit into the time unit. As such, we report arbitrary units of time in our simulation, as stated in previous revision rounds.

The value of the kinetic parameter L is rarely available due to challenges to calculating it from electronic structure or atomistic calculations or measuring it in experiment. In principle, one could design an experiment and perform a corresponding simulation to calibrate the L parameter value in real unit in the simulation to experiment. To do so is not an easy task and would require another major project, beyond the scope of the present work. Instead, in this work, we aim to address, not the value of the kinetic coefficient (or relative values between L_p and L_θ), but rather the role of mechanical clamping in setting the time scales for switching dynamics. In experiments, the L parameters are fixed by the material for both the freestanding and clamped cases. Therefore, in our simulation, we make an identical assumption, fixing kinetic coefficients and interrogating how the relative dynamics change upon changing mechanical boundary conditions. The time it takes to reach the final equilibrium state (of which there may be degenerate configurations that are equivalent, related by symmetry operations) will, of course, be dependent on choice of L (precisely why we report our simulation data in arbitrary units), but does not invalidate comparison across boundary conditions, which is the goal of the present work. Based on our work, both simulations and experiments indeed show that the main reason the free-standing membrane switches faster than the clamped film is because the barrier is significantly lower.

Reference

1. Tagantsev, A. K. "The role of the background dielectric susceptibility in uniaxial ferroelectrics." *Ferroelectrics* 69, no. 1 (1986): 321-323.
2. Tagantsev, Alexander K. "Landau expansion for ferroelectrics: Which variable to use?." *Ferroelectrics* 375, no. 1 (2008): 19-27.
3. Woo, C. H., and Yue Zheng. "Depolarization in modeling nano-scale ferroelectrics using the Landau free energy functional." *Applied Physics A* 91, no. 1 (2008): 59-63.
4. Levanyuk, Arkady P., Boris Anatolievich Strukov, and Andres Cano. "Background dielectric permittivity: Material constant or fitting parameter?." *Ferroelectrics* 503, no. 1 (2016): 94-103.
5. Rupprecht, G., and R. O. Bell. "Dielectric constant in paraelectric perovskites." *Physical Review* 135, no. 3A (1964): A748.
6. Schrade, D., R. Mueller, B. X. Xu, and D. Gross. "Domain evolution in ferroelectric materials: A continuum phase field model and finite element implementation." *Computer methods in applied mechanics and engineering* 196, no. 41-44 (2007): 4365-4374.
7. Su, Yu, and Chad M. Landis. "Continuum thermodynamics of ferroelectric domain evolution: Theory, finite element implementation, and application to domain wall pinning." *Journal of the Mechanics and Physics of Solids* 55, no. 2 (2007): 280-305.
8. Zhang, W., and K. Bhattacharya. "A computational model of ferroelectric domains. Part I: model formulation and domain switching." *Acta materialia* 53, no. 1 (2005): 185-198.
9. Chen, Long- Qing. "Phase- field method of phase transitions/domain structures in ferroelectric thin films: a review." *Journal of the American Ceramic Society* 91, no. 6 (2008): 1835-1844.
10. Ma, D. C., Yue Zheng, and C. H. Woo. "Phase-field simulation of domain structure for PbTiO₃/SrTiO₃ superlattices." *Acta materialia* 57, no. 16 (2009): 4736-4744.
11. Morozovska, Anna N., Sergei V. Kalinin, Eugene A. Eliseev, Venkatraman Gopalan, and Sergei V. Svechnikov. "Interaction of a 180 ferroelectric domain wall with a biased scanning probe microscopy tip: Effective wall geometry and

thermodynamics in Ginzburg-Landau-Devonshire theory." *Physical Review B* 78, no. 12 (2008): 125407.

12. Hlinka, J., and P. Marton. "Phenomenological model of a 90° domain wall in Ba Ti O 3-type ferroelectrics." *Physical Review B* 74, no. 10 (2006): 104104.

13. Li, Fei, Li Jin, Zhuo Xu, and Shujun Zhang. "Electrostrictive effect in ferroelectrics: An alternative approach to improve piezoelectricity." *Applied Physics Reviews* 1, no. 1 (2014): 011103.

14. Ikeda, Takuro. *Fundamentals of piezoelectricity*. Oxford university press, 1996.

REVIEWERS' COMMENTS

Reviewer #3 (Remarks to the Author):

I would like to thank the authors for their explanations. Below, I comment on individual answers of the authors only where necessary. The numbering refers to the numbering of the first, second and third review. From my point of view, only remark #9-iv deserves further action.

8i) Thank you for including the numerical value of the background dielectric constant in the paper.

8iii) Thank you for correcting the statement.

9-iv) Thank you for the explanations. I admit that my previous statement with regard to the use of different order parameters ("Most of the [...]") was not very helpful. In that regard, I agree with the authors that there are different phase-field models with different order parameters. Based on the discussion of the authors, also on the different notions of spontaneous polarization, I could now see a little better how they interpret their order parameter and its integration into their model. So this point is clarified. There is however one question that still puzzles me. When the authors apply a voltage of about 10 V and consider samples with a thickness of, say, 35 nm, this results in a maximum electric field of 2.86×10^8 V/m. Taking the electric permittivity of free space $\epsilon_0 = 8.854 \times 10^{-12}$ C/Vm and the background dielectric constant $\epsilon_b = 50$, the "background polarization" becomes $P_{\text{background}} = \epsilon_0 * (\epsilon_b - 1) * E = 8.854 \times 10^{-12} * 49 * 2.86 \times 10^8$ C/m² = 0.124 C/m². That value is in the range of the spontaneous polarization, which is about 0.6 C/m². That also becomes visible in the authors' plot in Figure R1. It is clear that the relation between "background electrostriction" and "spontaneous electrostriction" depends on the applied electric field. In their rebuttal, the authors have assumed a typical applied field of 500 kV/cm = 5×10^7 V/m, but in their paper they apply an electric voltage of 10 V, which results in higher electric fields for the films that they have considered. As they apply the electric loading in an instantaneous manner, the "background electrostriction" should be activated quickly in their simulations and therefore play a role. Do I misunderstand something? I would be grateful if the authors could explain once more.

18-v) While I agree that this all seems to be a matter of notation, I would like to emphasize that I did not term your model "nonlocal". I only raised the question if the authors are having a nonlocal model in mind and received the answer "[...] As such, we should use F_{tot} , i.e., the total energy, and not the local value, f . The referee is correct that this is a non-local theory as we are minimizing the energy of the complete system." (see response to review 2). Now, according to your new definition (18.1) you are indeed using the local energy, which is fine, so this question is answered.

18-vi) While I would still think that the simulation results will depend on the mentioned parameters and their relation, I understand that it will not be straightforward for the authors to determine them at present. In that regard, I will not insist on any further discussion and accept the answer of the authors as is.

Point-by-point response to the reviewers' comments

Summary of changes

We appreciate the reviewer's questions and comments. Below are our detailed responses to each of the reviewer's comments.

Reviewer #3 (Remarks to the Author):

I would like to thank the authors for their explanations. Below, I comment on individual answers of the authors only where necessary. The numbering refers to the numbering of the first, second and third review. From my point of view, only remark #9-iv deserves further action.

8i) Thank you for including the numerical value of the background dielectric constant in the paper.

8iii) Thank you for correcting the statement.

9-iv) Thank you for the explanations. I admit that my previous statement with regard to the use of different order parameters ("Most of the [...]") was not very helpful. In that regard, I agree with the authors that there are different phase-field models with different order parameters. Based on the discussion of the authors, also on the different notions of spontaneous polarization, I could now see a little better how they interpret their order parameter and its integration into their model. So this point is clarified. There is however one question that still puzzles me. When the authors apply a voltage of about 10 V and consider samples with a thickness of, say, 35 nm, this results in a maximum electric field of $2.86 \cdot 10^8$ V/m. Taking the electric permittivity of free space $\epsilon_0 = 8.854 \cdot 10^{-12}$ C/Vm and the background dielectric constant $\epsilon_b = 50$, the "background polarization" becomes $P_{\text{background}} = \epsilon_0 \cdot (\epsilon_b - 1) \cdot E = 8.854 \cdot 10^{-12} \cdot 49 \cdot 2.86 \cdot 10^8$ C/m² = 0.124 C/m². That value is in the range of the spontaneous polarization, which is about 0.6 C/m². That also becomes visible in the authors' plot in Figure R1. It is clear that the relation between "background electrostriction" and "spontaneous electrostriction" depends on the applied electric field. In their rebuttal, the authors have assumed a typical applied field of 500 kV/cm = $5 \cdot 10^7$ V/m, but in their paper they apply an electric voltage of 10 V, which results in higher electric fields for the films that they have considered. As they apply the electric loading in an instantaneous manner, the "background electrostriction" should be activated quickly in their simulations and therefore play a role. Do I misunderstand something? I would be grateful if the authors could explain once more.

Response: We thank the referee for the comment. Before we get into the details of the role of the applied field and the background dielectric constant, we would like to point out that the primary focus of our paper is to ask the question: what happens to the lattice strain that occurs DURING the switching of the polarization, for example from the UP state to the DOWN state. In the case of BFO, with a spontaneous polarization of 0.9C/m^2 , the switching of the spontaneous polarization from the UP to the DOWN state occurs via the rotation of the dipole moment through a sequence of steps (Heron, Nature 2014). The strong coupling between the spontaneous electric dipole and the lattice distortion therefore means that DURING the switching of the polarization, there is a time-dependent, dynamic strain that arises in the film. In the case of the free-standing film, this strain is not impeded, unlike in the case of a film that is tethered to a substrate. In the case of such a film tethered to a substrate, the time-dependent strain during switching, in turn, leads to a stress in the film that acts like a mechanical impediment to the switching process, thus increasing the switching voltage (which is a proxy for the switching energy) as well as its time dynamics. That is the essence of our paper.

Now, coming to the phase field calculations and the role of the background dielectric constant and the induced polarization due to the applied field of $10\text{V}/35\text{nm}$. As the referee points out, this leads to an induced polarization of the order of $\sim 0.1\text{C/m}$. This induced polarization is $\sim 10\%$ of the spontaneous polarization; more importantly, at the high fields of $10\text{V}/35\text{nm}$, this induced polarization will lead to an induced strain that is of the order of a few % of the spontaneous strain (within the simplistic model of the strain, $x = QP^2$ (M. E. Lines and A. M. Glass, *Principles and Applications of Ferroelectrics and Related Materials*, 2001), where Q is the electrostriction coefficient and P is the polarization). When P is set to the spontaneous polarization: $x = Q \cdot 0.9^2$; for the case of the induced polarization of 0.1C/m^2 , the induced strain is : $x = Q(0.1)^2$. Therefore, the induced polarization (at a high field of $2.85 \times 10^6\text{V/cm}$) contributes a small fraction of the spontaneous strain. Thus, we believe that the induced polarization due to the applied field, is a correction to the spontaneous polarization and does not change the broader conclusions of this paper in any substantive way. Furthermore, as stated in the first paragraph, we strongly believe that this influence of the induced strain is not particularly relevant to the problem statement on hand. The induced strain is constant for a constant applied field, while we are interested in the dynamic evolution of the strain as the spontaneous polarization is being switched. Finally, we also note that the induced strain arising from the background dielectric constant is essentially the same for the free-standing and clamped films; since we are interested in the difference between these two scenarios, the relevant strain is the dynamic strain originating from the switching of the spontaneous polarization.

18-v) While I agree that this all seems to be a matter of notation, I would like to emphasize that I did not term your model "nonlocal". I only raised the question if the authors are having a nonlocal model in mind and received the answer "[...] As such, we should use F_{tot} , i.e., the total energy, and not the local value, f . The referee is correct that this is a non-local theory as we are minimizing the energy of the complete system." (see response to review 2). Now, according to your new definition (18.1) you are indeed using the local energy, which is fine, so this question is answered.

18-vi) While I would still think that the simulation results will depend on the mentioned parameters and their relation, I understand that it will not be straightforward for the authors to determine them at present. In that regard, I will not insist on any further discussion and accept the answer of the authors as is.